# Mechanisms Underlying the Anti-Depressive Effects of Regular Tea Consumption

**DOI:** 10.3390/nu11061361

**Published:** 2019-06-17

**Authors:** Dylan O’Neill Rothenberg, Lingyun Zhang

**Affiliations:** Department of Tea Science, College of Horticulture Science, South China Agricultural University, Guangzhou 510640, China; Dylan.Rothenberg@colorado.edu

**Keywords:** *Camellia sinensis*, depression, inflammation, HPA axis, gut–brain axis, neurogenesis, neurotransmission, SCFA, EGCG, theaflavin, L-theanine

## Abstract

This article is a comprehensive review of the literature pertaining to the antidepressant effects and mechanisms of regular tea consumption. Meta-data supplemented with recent observational studies were first analyzed to assess the association between tea consumption and depression risk. The literature reported risk ratios (RR) were 0.69 with 95% confidence intervals of 0.62–0.77. Next, we thoroughly reviewed human trials, mouse models, and in vitro experiments to determine the predominant mechanisms underlying the observed linear relationship between tea consumption and reduced risk of depression. Current theories on the neurobiology of depression were utilized to map tea-mediated mechanisms of antidepressant activity onto an integrated framework of depression pathology. The major nodes within the network framework of depression included hypothalamic-pituitary-adrenal (HPA) axis hyperactivity, inflammation, weakened monoaminergic systems, reduced neurogenesis/neuroplasticity, and poor microbiome diversity affecting the gut–brain axis. We detailed how each node has subsystems within them, including signaling pathways, specific target proteins, or transporters that interface with compounds in tea, mediating their antidepressant effects. A major pathway was found to be the ERK/CREB/BDNF signaling pathway, up-regulated by a number of compounds in tea including teasaponin, L-theanine, EGCG and combinations of tea catechins and their metabolites. Black tea theaflavins and EGCG are potent anti-inflammatory agents via down-regulation of NF-κB signaling. Multiple compounds in tea are effective modulators of dopaminergic activity and the gut–brain axis. Taken together, our findings show that constituents found in all major tea types, predominantly L-theanine, polyphenols and polyphenol metabolites, are capable of functioning through multiple pathways simultaneously to collectively reduce the risk of depression.

## 1. Introduction

Depression is the most common mental health condition in the general population [1,2], characterized by despondency, loss of motivation or pleasure, feelings of guilt or low self-esteem, poor appetite, insomnia, fatigue, and difficulty concentrating [3]. In extreme cases, depression can lead to suicide [4,5], and increase the risk of morbidity and all-risk mortality [6]. Depression can become an all-consuming disorder that affects occupational potential [7] and quality of life [8,9]. The World Health Organization (WHO) predicts that, by 2020, depression will rank second in global disease burdens and one of the priority conditions covered by the WHO’s Mental Health Gap Action Programme [10].

Progress in the treatment of major depressive disorder (MDD) has been modest at best, in part because of the withdrawal of much of the pharmaceutical industry from the development of new psychotropic drugs [11]. Average estimates calculate that, among patients entering treatment for major depressive disorder (MDD), only about one third enter remission four months later [12]. With rates of MDD rising and treatments and efficacy of MDD treatments remaining stagnant, it is time to start looking elsewhere for ways to treat MDD. Some data have been encouraging for the ability of lifestyle changes to reduce risk of MDD such as less smoking [13], less alcohol consumption [14], fewer high-fat foods [15], and more exercise [16]. It has also been found that regular tea consumption has a strong linear relationship with reduced MDD [17]. This has led researchers to look more deeply into the mechanisms underlying the use of tea as an antidepressant agent. 

Tea is the second most consumed beverage after water, mostly because it is an affordable luxury consumed by all socioeconomic classes around the world. As such, even minor benefits to mental or physical health could carry considerable implications for public health. Tea has been well established to confer a number of benefits, including anti-diabetic, anti-aging, pro-cardiovascular, and pro-metabolic effects [18,19,20]. The mechanisms of tea phytochemicals in reducing the risk of depression are complex and multidimensional, since neurologists and psychiatrists are coming to realize that depression itself is a multifaceted neurobiological pathology. 

However, our understanding of depression is maturing, and new theories are emerging that allow us to see its deeper, more fundamental causes. Most data on the topic reflect that tea consumption can reduce risk of depression, but to date, no studies have used modern theories on depression pathology to provide comprehensive analysis of the mechanisms underlying these observations. Therefore, this report uses a current conceptual framework of depression neurobiology to provide detailed insight on the mechanisms underlying the observed antidepressant effects of tea consumption. 

## 2. Concept of Tea within an Integrative Theory of Depression

Neuroimaging data have provided scientists with insights that have elucidated our understanding of the human brain. These insights have led to the development of an integrated theory of depression, in which multiple dysfunctional systems are viewed as a single interconnected pathology [21,22,23]. In the instance of an outdated theoretical model, for example the monoamine hypothesis, low monoamine levels (typically serotonin) is the cause of depression, and the way to treat the illness is by increasing circulating levels of serotonin. This treatment methodology has proven to be far too reductionist, and that is reflected in low efficacy rates of selective serotonin reuptake inhibitors (SSRIs) in long-term MDD treatment. The monoamine hypothesis fails to address what is responsible for the upstream causes of initial dysfunction within the serotonergic system, and thus simply supplying more serotonin has been shown to be an effective short-term alleviation of depressive symptoms, but not a long-term solution to the pathology.

A unified theory of depression, sometimes referred to as an integrated theory of depression, is one that underscores the interconnectivity among neurobiological systems involved in depression pathology, including inflammation, hypothalamic–pituitary–adrenal (HPA) axis activity, neurogenesis/neuroplasticity, and monoaminergic systems, including the gut–brain axis [23]. This article uses an integrative theory of depression pathology to explain the multitude of ways in which compounds in tea may be exerting anti-depressive effects in humans. Conceptually speaking, individual compounds in tea are not exerting potent pharmacological effects on individual neurobiological systems, as SSRIs do on the serotonergic system, but rather multiple constituents in tea, including L-theanine, L-arginine, polyphenols and their metabolites are collectively and simultaneously exerting modest effects within multiple neurobiological systems, leading to an overall significant net risk reduction in depression.

Thus, the primary focus of this paper is to review research detailing the mechanistic actions of tea compounds within neurobiological systems pertaining to depression, compare and contrast those findings, and map their functional significance within the larger network of integrated depression pathology. 

## 3. Antidepressant Effects of Tea Consumption in the Literature

In accordance with rigorous standards of research quality, 11 studies with 13 reports were selected for meta-analysis that evaluated the association between tea consumption and depression risk [17]. Meta-data were comprised of five cohort reports and eight cross-sectional reports, all of which yielded at least moderate quality assessment scores, averaged at good within Newcastle–Ottawa and Agency for Healthcare Research and Quality assessment scales. Studies included 22,817 participants with 4743 cases of depression, and a dose–response analysis of 10,600 participants with 2107 cases. This meta-analysis found that higher consumption of tea was associated with lower risk of depression. Dose–response analysis identified a negative correlation between tea consumption and depression risk, with every three cups/day increment in tea consumption associated with a 37% decrease in the risk [17]. A subgroup analysis was conducted to assess whether lifestyle factors such as physical exercise, smoking, alcohol consumption, etc. were confounding the results; however, similar results were obtained in these subgroup analyses. The aggregated risk ratio (RR) for studies measuring green tea (*n* = 3) was comparable to studies measuring “diverse tea types” (*n* = 5), which included oolong, black, white, and pu-erh teas, with RR (95% confidence interval (CI)) of 0.67 (0.56–0.79) and 0.69 (0.62–0.77) for green and diverse tea types, respectively. 

In the meta-analysis of association in the 13 reports, all but one subgroup found similar results regarding associative risk. The inconsistent finding came from the population-based Kuopio Ischaemic Heart Disease Risk Factor Study taken between 1984 and 1989 and followed until the end of 2006, which investigated the association between intake of coffee, tea and caffeine and severe depression in middle-aged Finnish men [24]. No association was observed between severe depression and intake of tea in this study. However, this prospective study should be interpreted with caution, since it focused only on severe depression based on discharge diagnosis, which may have overlooked cases of mild and moderate depression in which patients were not hospitalized. Therefore, it is possible that this study was biased towards a null hypothesis. The meta-analysis by Dong et al. offered a brief mention of feasible mechanistic explanations for the results. However, it did not take into consideration comprehensive integrated views of depression pathology, perhaps because these theories on depression have only emerged in very recent years [21,22,23].

We conducted another search for high quality data measuring tea consumption and depression risk to supplement the meta-analytical data from Dong et al. [17] with more recent research. The MEDIS study published in May 2018 enrolled 2718 older individuals from 22 Mediterranean islands in cross-sectional sampling in the period of 2005–2011. A broad ranging set of dietary habits and socio-demographic characteristics were analyzed through cross-sectional examination for associations with depression [25]. Diet-related factors included consumption quantities and frequencies of fish, meat, vegetables, legumes, coffee, tea, and various alcoholic beverages. Factors such as age, education, financial status, physical activity, various blood lipid parameters and BMI were measured. Logistic regression model evaluating the various factors associated with depression found that daily tea consumption showed the lowest RR of any metric measured in the study (RR: 0.51; 95% CI; 0.40–0.65, *p* value < 0.001). The MEDIS study provided no detailed mechanistic explanation for the observed risk reduction resulting from daily tea consumption [25]. 

Using data from the Korean National Health and Nutrition Examination Survey, a total of 9576 (3852 men and 5725 women) aged 19 years or older were cross examined for associations between green tea consumption and self-reported depression [26]. Consumers of more than three cups/week had 21% lower prevalence of depression (RR = 0.79, 95% CI = 0.63–0.99, *p* value = 0.0101) after adjusting for confounding factors. A weakness of this study was that the highest consumption bracket reported was three cups/week, when in fact several studies found significant differences in risk reduction between consuming three cups/week, considered moderate consumption, and one or more cups/day, considered high consumption. If this study had stratified between moderate and high consumption groups, it is possible there would have been lower RR for high consumption groups, as was the case in meta-data by Dong et al. [17]. 

Seventy-four healthy subjects participated in a double-blind, randomized placebo-controlled study with oral administration of green tea or placebo for five weeks, after which measurements were taken that included reaction time of reward responsiveness, and scores on the Montgomery–Asberg depression rating scale (MADRS) and 17-item Hamilton Rating Scale for Depression (HRSD-17) to estimate the depressive symptoms in these two groups. The results show chronic treatment of green tea increased reward learning compared with placebo by decreasing the reaction time in monetary incentive delay task. Moreover, participants treated with green tea showed reduced scores measured in MADRS and HRSD-17 compared with participants treated with placebo [27]. 

Taken together, these data suggest that regular daily tea consumption contribute to risk reduction in healthy individuals, and evidence was particularly strong among aging populations. 

## 4. Mechanistic Considerations

Four primary nodes are named as mechanistic systems, which interrelate within a network of depression pathology: the HPA axis; immune inflammatory response; monoaminergic systems, which includes the gut–brain axis; and neurogenesis/neuroplasticity (Figure 1). These four nodes each have important subsystems within them, including signaling pathways, specific target proteins, or transporters that interface with compounds in tea, mediating their antidepressant effects. The healthy functioning of each node has the potential to be compromised by genetic or biological factors, representing poor “hardware” of these nodes, or by environmental stressors, learned behaviors, psychosocial stress, etc., representing poor “software” of this system [28]. Regardless of the upstream cause, when compromised, these nodes are able to reduce the functional capacity of other nodes through the formation of harmful positive feedbac k loops, creating an interrelated network of neuro-pathologies that can predispose MDD. Thus, the alleviation of depressive pathology within any single node has the capacity to reverberate through to other nodes, and similarly, targeting multiple nodes simultaneously may translate to significant risk reductions, which we hypothesize is the mode of action underlying the observed antidepressant effects of tea consumption. We call this the “reduce and restore” hypothesis, and the network nodal targets include reduction of HPA axis hyperactivity; reduction of inflammation; restoration of monoaminergic systems, including restoration of neurologically active gut microbiota; and restoration of neurogenesis/neuroplasticity. 

### 4.1. Reduction of HPA Axis Hyperactivity 

The hypothalamic–pituitary–adrenal (HPA) axis is a feedback loop in which stress activates a hormonal pathway that induces a “fight-or-flight” response to a perceived threat. Due to its stress-induced nature, the HPA axis is referred to as the “stress circuit”. Mechanistically speaking, when mammals experience stress, the hypothalamus releases corticotropin-releasing hormone (CRH). CRH then acts on the pituitary gland to trigger the release of adrenocorticotropin (ACTH) into the bloodstream. In response to CRH, ACTH acts on the adrenal cortex by binding to the adrenal melanocortin 2 receptor (MC2R), which signals through cyclic AMP to stimulate the production and secretion of glucocorticoids (CORT). In humans, the primary CORT is cortisol, or known more colloquially as the “stress hormone”. Two of cortisol’s primary functions are promotion of gluconeogenesis in order to increase blood glucose levels, and the suppression of immune function. The HPA axis has considerable impact on an individual’s mood at any given time, as this system communicates readily with other neurocircuitry in the brain, such as the limbic system, which serves to regulate mood and motivation. 

Within a typical HPA axis response, the acute stressor will pass, and stress hormone cortisol will exert a negative feedback effect on the stress circuit by down-regulating CRH synthesis in the hypothalamus. However, unresponsiveness to cortisol’s negative feedback signaling, or periods of constant perceived stress will cause the stress circuit to operate continuously, a physiological state referred to as HPA axis hyperactivity. Under such conditions, the limbic system is over-stimulated, leading to a condition of “learned helplessness”, in which the overwhelmed subject loses motivation and detaches from external engagement, at which point in humans, depressive symptoms are reported and MDD is often diagnosed. Furthermore, with HPA hyperactivity, the hippocampus becomes desensitized to constant elevated levels of CRH, causing the body to release more cortisol in an attempt to down-regulate CRH production. The elevated cortisol causes more stress and inflammation, in turn triggering the release of more cortisol, and the stress circuit becomes a positive feedback loop of neuropathology (Figure 2). Over time, chronic elevated stress hormone levels deteriorate important neural networks in the brain, including monoaminergic and limbic systems, and reduce hippocampal volume and neuroplasticity. As discussed below, these downstream effects can further exacerbate MDD neuropathology and symptomology. For these reasons, some have described MDD to be nearly synonymous with dysfunctional HPA axis activity [29,30]. This section reviews in vitro and in vivo studies measuring how the bioactive constituents in tea have been observed to affect HPA axis activity under stressful conditions, including the mechanistic explanations proposed for the observed effects (summarized in Table 1). 

A recent study intraperitoneally injected ICR mice with L-theanine (20 mg/kg) for seven days before a single intraperitoneal injection of lipopolysaccharide (LPS) to induce an inflammatory stress response. The LPS treatment caused a significant increase in circulating ACTH and CORT, and pretreatment with L-theanine significantly attenuated this response. The proposed mechanism was a modulatory interaction between theanine and glucocorticoid receptors (GR), although the immediate regulating action of theanine on GR was said to be unknown [33]. 

Wang et al. [32] conducted another experiment similarly testing the effects of L-theanine on HPA activity, however theanine was administered intragastrically, and stress was induced in mice by whole body heat shock rather than LPS. Heat shock is a stress-induction method that has been shown to activate HPA activity and increase CORT levels in previous in vivo experiments [45]. L-theanine supplementation normalized the heat stress-induced HPA axis hyperactivity by significantly reducing elevated plasma ACTH and CORT levels compared to control. Wang et al. [32] suggested that theanine might have been able to participate in crosstalk between inflammatory cytokines and the HPA axis, but commented that the immediate mechanism of HPA regulatory activity was uncertain.

Some studies have suggested that a potential mechanism by which L-theanine can normalize HPA hyperactivity is through the antagonistic effect on glutamate. Glutamate is the primary excitatory neurotransmitter in the central nervous system, critical to HPA axis activation [46], reflected in studies where glutamate agonists activated the HPA axis in rats by inducing ACTH elevation [47]. L-theanine incorporated into the brain is reported to reduce the release of glutamate from pre-synapse to the synaptic cleft [48], where glutamate can then be decarboxylized into gamma-aminobutyric acid (GABA), the brain’s main inhibitory neurotransmitter. In support of this mechanistic explanation, one study measured the hippocampus of mice that ingested theanine (6 mg/kg) in drinking water for two weeks, finding that the levels of glutamate and pyroglutamic acid were significantly reduced, and GABA increased [49], suggesting that theanine modulated GABA production via glutamate inhibition. This proposed pathway would provide stress reduction benefits that are twofold: reducing HPA axis activation through reduction of activator glutamate, and increasing the circulating levels of GABA, which has been shown to provide anti-stress/anxiolytic effects. It remains debatable whether GABA synthesized outside the brain is able to cross the blood–brain barrier (BBB) in quantities with relevance to in cerebro neurotransmission [50], which suggests that up-regulation of GABA synthesis in the brain could be an important contribution of L-theanine to lowering stress and normalizing HPA axis activity.

Green tea polyphenols (GTP) without L-theanine orally administered in doses of 5, 10 and 20 mg/kg for seven days showed antidepressant-like effects in a mouse model of depression [34]. Following GTP supplementation, serum corticosterone, the primary CORT in rodents, and ACTH levels were reduced in mice exposed to forced swimming tests (FST). GTP also significantly reduced immobility in FST and tail suspension tests (TST), suggesting that tea polyphenols may possess the ability to normalize stress-induced HPA activity and provide antidepressant effects through this pathway.

Both L-theanine and tea polyphenols have been shown capable of attenuating HPA-mediated responses when ingested individually, however one study showed that these compounds acted antagonistically regarding their effects on stress response. Unno et al. [31] measured stress responses in mice following consumption of three types of green tea in a model of psychosocial stress induced by confrontational housing. Adrenal hypertrophy was used as a biomarker for stress response, which is common among studies measuring HPA activity in living organisms [51,52]. This study found that two tea types, theanine-rich Gyokuro and low-caffeine Sencha, significantly reduced adrenal hypertrophy in stressed mice, while the third tea type, normal Sencha, reduced stress levels at an insignificant rate compared to the water control. The reason for this effect was discovered to be that not only caffeine, but also epigallocatechin gallate (EGCG) were both effective at suppressing anti-stress effects of L-theanine. Interestingly, epigallocatechin (EGC), the non-gallate form of EGCG, was found to block the antagonistic effects of EGCG against L-theanine, effectively preserving the anti-stress effects of theanine. EGC was able to do this through competitive inhibition of EGCG uptake into the brain. A competitive inhibitory relationship was assumed as the blood-brain permeability of EGCG was significantly lowered by the same molar amount of EGC using a BBB model. By lowering the incorporation of EGCG into the brain, EGC lowered the antagonistic effects of EGCG against L-theanine. EGC itself, however, was not shown to inhibit the anti-stress effects of L-theanine. This may be because efflux transporters expressed on the cell membrane have been shown to be actively suppressed by the presence of a gallate moiety [53,54,55], meaning that EGC effluxed more easily out of cell membranes, and was thus less able to interfere with L-theanine. 

No studies to date have proposed a mechanistic explanation of how EGCG may have interfered with the anti-stress effects of L-theanine; however, we propose here an original hypothesis for this observed effect. It has been suggested that L-theanine has another pathway of conferring anti-stress effects in addition to the pathway involving decarboxylation of glutamate into GABA. It was suggested by Yamada et al. that theanine was able to act directly on α-amino-3-hydroxy-5-methyl-4-isoxazolepropionic acid (AMPA) receptors, ionotropic glutamate receptors in the brain, and in rats, action upon AMPA receptors by L-theanine lead to significantly increased glycine concentration in the striatal interstitium [56]. Glycine is a major inhibitory neurotransmitter in the spinal cord and brainstem responsible for processing motor and sensory information, functioning in the CNS similarly to, and simultaneously with GABA [57,58]. In addition to the anxiolytic effects normally attributed to inhibitory neurotransmitters, glycine also induces the release of dopamine (DA) [59], which may explain at least part of the reported mood-enhancing effects of L-theanine [60]. In the pathway described above, L-theanine acts on AMPA receptors, causing the release of glycine, which subsequently acts on glycine receptors, triggering the release dopamine. However, it was shown that EGCG significantly modulated AMPA receptor activity by inhibiting calcium influx into the neuronal cell [37]. The effect of EGCG on AMPA receptors were similar to CNQX, a competitive antagonist of AMPA receptors [61]. In these two studies, the overall effect of EGCG on AMPA receptors was neuroprotective in nature, as EGCG effectively prevented excitotoxicity induced by high concentrations of an AMPA agonist. However, we hypothesize that EGCG may be competing with L-theanine for AMPA receptor binding sites, or through some other mechanism impeding its affinity for AMPA receptor activity, and through this mechanism lowering the total net anti-stress effect conferred by theanine. Thus, our original “AMPA hypothesis” posits that gallated polyphenols in tea may be capable of reducing anti-stress effects of L-theanine via competitive inhibition of AMPA receptor binding sites. Thus far, EGCG is the only gallated tea polyphenol with data regarding its AMPA receptor-modulating potential or capacity to inhibit anti-stress effects of L-theanine, however other gallated tea polyphenols such as ECG, EGCG3”Me, or theaflavin-3,3′-digallate may serve as interesting targets of future research regarding said effects. 

The antagonistic effect of caffeine against theanine-induced stress reduction was tested in humans using green tea with lowered caffeine content (LCGT) in a group of middle-aged individuals (*n* = 20, mean age 51.3 ± 6.7 years) in a double-blind crossover design with standard green tea (SGT) as a control [41]. A non-invasive biomarker for stress was utilized, which measured the body’s alternate system for mediating stress response other than the HPA axis, the sympathetic branch of the autonomic nervous system (ANS). Salivary α-amylase (sAA), a digestive enzyme in the mouth, has been shown to be a useful tool in measuring ANS reactivity to stress [62] because it rapidly increases in response to physiological and psychosocial stress [63,64,65]. The results of the human trial showed that sAA levels were considerably lower in the participants who consumed LCGT compared to those that consumed SGT, suggesting that caffeine reduced anti-stress effects of L-theanine. This study also found that sleep quality improved significantly with LCGT compared to SGT, which is relevant to depression, as sleep disturbances have been shown to act a precursor to the development of depression [66]. This finding corroborates with other research reporting the ability of L-theanine to improve sleep duration and sleep quality, particularly when ingested together with GABA, an amino acid found in high quantities in teas processed under certain conditions [67,68]. Unno et al. [41] elucidated the importance of low relative caffeine content in mediating the ability of tea consumption to improve sleep and reduce bodily systemic responses to everyday stress. It is worthwhile to note, however, that, although caffeine may reduce anti-stress effects of tea, some studies show that certain tea compounds, such as polyphenols, theobromine and L-theanine, can enhance mood and cognitive effects of caffeine and alleviate negative psychophysiological effects of caffeine [60], which may confer added benefits to university students, among whom 92% reported to have consumed caffeine in the past year in a geographically-dispersed sample of United States university students [69]. 

Reduced sAA levels (*p* = 0.032) and lower subjective stress levels (*p* = 0.020) were reported among fifth-year university students (*n* = 20) who consumed 200 mg of theanine twice per day [70]. Similarly, green and white tea consumption decreased levels of salivary chromogranin A (CgA), an ANS-mediated indicator of stress level, among university students (*n* = 18) performing mental stress load tasks in a randomized cross-over design study [42]. CgA was also found to decrease in two studies that exposed university students to the aromas of black or green tea prior to taking 30-min mental stress load tasks [43,44]. The inhalation of both types of tea aroma induced lower CgA levels following stress load tasks. 

Zhao et al. [35] found that administration of EGCG or a GTP mixture was associated with partial restoration of normal plasma CORT levels in mice following four-week restraint stress-induced HPA hyperactivity. The primary mechanism of CORT reduction was said to be upregulation of extracellular signal-regulated kinase (ERK)1/2 signaling pathways, which was induced significantly by EGCG treatment and to a lesser extent by GTP [35]. This is a fascinating finding in light of increasing evidence suggesting that ERK pathway suppression may be critically linked to the development of depression [71,72]. It was shown that ERK1/2 is highly sensitive to stress and closely related to cognition and mood processing [73]. Moreover, ERK1/2 was found to be significantly reduced in post-mortem brains of suicide victims [74]. Multiple data have shown that EGCG and GTP mixtures can inhibit HPA activity and improve neurological status in mice via activation of the ERK pathway [35,75]. However, it remains unclear if the anti-depressive effects of ERK are predominantly due to ERK itself, or by the phosphorylation of its downstream substrate, cyclic AMP response element binding protein (CREB), or by the consequent upregulation of its further downstream target, brain-derived neurotrophic factor (BDNF). BDNF is a key promoter of hippocampal neurogenesis, and represents another target in combating MDD, which is discussed in more detail when reviewing the role of tea compounds in neurogenesis and neuroplasticity. 

In corroboration with the findings of ERK activation by EGCG and GTP, a recent study found that intraperitoneally injected EGCG (25 mg/kg) for 14 days prior to a single prolonged stress (SPS) was able to significantly decrease the SPS-stimulated increase in plasma corticosterone levels, and significantly attenuate SPS-induced increases in plasma CRH and ACTH levels. These changes were accompanied by attenuated CREB/BDNF levels, and attenuation of ameliorated cognitive ability and object recognition memory in behavioral tests. This study provides evidence of an up-regulated ERK/CREB/BDNF pathway working in unison with normalized HPA axis activity to mediate antidepressant activity of EGCG [36]. 

Johnston proposed a hypothesis for the mechanism of EGCG’s anxiolytic effect that included direct binding to the benzodiazepine binding site of GABA receptor complexes [76]. In doing so, EGCG may be able to displace endogenous negative modulators of the GABA receptor. In theory, this makes sense, as the presence of endogenous negative modulators of the benzodiazepine binding site of GABA receptors have been found to exist in rats [77]. These polypeptide compounds possessed diazepam-binding inhibition (DBI) properties in the hippocampi of rats, where they elicited “proconflict” responses. EGCG could be working to restore full GABA receptor function by the displacement of DBI-type anxiogenic ligands [76].

As mentioned above, EGCG was able to antagonize AMPA glutamate receptors in mice, decreasing neuronal excitotoxicity through that mechanism [37], albeit that may have lowered total anti-stress effects by interfering with L-theanine. Neuroprotective effects via attenuated over-excitation in neurons has been reported for EGCG in several studies [38,78]. Hypothalamic infusion of EGCG in rats [31] lowered glutamate and norepinephrine levels, and increased GABA levels in the paraventricular nucleus (PVN), indicating attenuated ANS stress-response, coinciding with a more balanced environment of circulating excitatory and inhibitory neurotransmitters [39]. 

The SIRT1/PGC-1α signaling pathway was proposed as another mechanism through which EGCG was able to attenuate stress-response via the normalization of excitatory and inhibitory neurotransmitters [40]. PGC-1α (peroxisome proliferator-activated receptor gamma (PPARγ) coactivator-1α) is a key regulator of cellular energy metabolism in the brain, found to be markedly declined in stressed animals, and significantly attenuated by EGCG and GTP treatments [35,40,79]. PGC-1α regulates the expression of transcriptional factors involved in energy metabolism and inflammation in high energy demand tissues, such as the cerebrum. In recent studies on mice, PGC-1α activation was found to mediate improvements in stress management [80] and reduction of depressive behavior [81]. The anxiolytic effects of PGC-1α are due to its role in GABA neurotransmission, wherein deletion of PGC-1α was found to promote asynchronous GABA release [82,83]. For these reasons, the EGCG- and GTP-induced enhancement of PGC-1α expression in the brain of stressed animals may be critical to mediating their effects on neurotransmission, stress and depression. 

Taken together, preliminary results appear to be promising regarding the effects of tea compounds on dysregulated stress-response systems, although more data are necessary to understand their effects in humans, particularly given the sensitivity of L-theanine’s effects to caffeine and EGCG. Notable among the stress-response normalization pathways of tea are glutamatergic antagonism, GABAergic pathway protection, and attenuation of ERK1/2 signaling and PGC-1α signaling. However, several studies reported that the mechanisms underlying attenuated stress-responses are unknown, highlighting the need for more research in this field. 

### 4.2. Reducing Inflammation 

Inflammation is triggered by the release of inflammatory cytokines such as interleukin (IL)-1β IL-2, IL-6, tumor necrosis factor alpha (TNF-α), and C-reactive protein (CRP), which all have been found to be up-regulated in patients with depression [84,85]. The relationship between inflammation and depression can be traced in part back to an adaptive trait called “sickness behavior”, in which humans who are attempting to fight infection reduce participation in activities such as exploration or seeking mates, which compete with an active immune system for energy stores [86]. For an acute infection during evolutionary times, behaviors of anhedonia, fatigue, and internal focus allowed one to allocate bodily resources to the immune system, which can consume 30% of basal metabolic rate during an immune response. However, in cases of chronic inflammation, which can come as a result of poor sleep, poor diet, smoking, lack of exercise, psychological stress, or a number of other lifestyle factors, sickness behavior can persist indefinitely, no longer serving its evolutionary purpose [87]. 

Additionally, inflammation coincides with a significant decrease in serotonin levels. The mechanistic explanation is that serotonin synthesis and bacterial growth both depend on tryptophan as a precursor. The immune system attempts to arrest bacterial growth by up-regulating a metabolic pathway that metabolizes tryptophan into kynurenine (KYN) via the enzyme indoleamine 2,3-dioxygenase (IDO). Inflammatory cytokines induce IDO activity, causing a significant drop in tryptophan levels and a corresponding decline in serotonin synthesis. The KYN pathway might not necessarily be a part of sickness behavior, however low serotonin levels resulting from inflammation almost certainly exacerbate depression symptomology. 

Finally, inflammatory cytokines can boost cortisol levels and activate the HPA axis, which is in turn inflammatory. As discussed in the previous section, inflammation can form a positive feedback loop with the HPA axis, creating a state of chronic stress and inflammation, leading to the deterioration of important neural networks in the brain, such as the monoaminergic and limbic systems, and hippocampal neuroplasticity. For these reasons, reducing inflammation has become a critical target in the treatment of depression [11,88,89,90]. Here, we review in vitro and in vivo studies that have measured the effect on inflammation of ingesting tea or individual bioactive compounds in tea, including the mechanistic explanations proposed for the observed effects (summarized in Table 2). 

In mice with LPS-induced neural inflammation, black tea theaflavins (TF) suppressed production of inflammatory cytokines and significantly reduced depression-like behavior, as measured by reduced immobility in the TST [91]. Interestingly, this study compared the anti-inflammatory activity of TF to that of catechin, EGCG, chlorogenic acid, and caffeic acid at concentrations of 0, 1, 3, 10, and 30 μM. The results indicate that the anti-inflammatory activities of TF on microglia are stronger than those of common polyphenols, but comparable to EGCG. In two more in vivo studies, theaflavin-3,3′-digallate (TF3) significantly inhibited LPS-induced expression of several inflammatory biomarkers, including TNF-α, IL-1β, and IL-6 [92,93]. These data indicate that black tea TFs may be effective anti-inflammatory agents with comparable efficacy to EGCG. 

In a high-fat diet (HFD) mouse model, green tea extract (GTE) treatment reduced hepatic inflammation by decreasing pro-inflammatory signaling through TNF receptor-1 and Toll-like receptor (TLR)-4, which in non-GTE HFD mice increased nuclear factor kappa-light-chain-enhancer of activated B cells (NFκB) activation, and consequent liver injury [94]. NF-κB is a protein complex involved in cellular responses to stress and inflammation, and controls transcription of DNA, cytokine production and cell survival. The NF-κB pathway, in addition to another potentially inflammatory pathway, the mitogen-activated protein kinase (MAPK) pathway, were found to be down-regulated following EGCG supplementation in a high-fat/fructose diet (HFFD) mouse model [95]. Similarly, teasaponin, another major active component in tea, delivered orally (0.5%) mixed in HFD for eight weeks was found to attenuate inflammatory markers in mice hippocampi, including TLR-4, NF-κB, IL-1β, IL-6 and TNF-α [105]. These data suggest that several active compounds in tea may be effective in attenuating inflammation resulting from high-fat diets. 

In corroboration with these findings, Lee et al. [36] found that intraperitoneally injected EGCG (25 mg/kg) for 14 days prior to a single prolonged stress (SPS) was able to decrease the SPS-induced increases of IL-1β and TNF-α levels in the hippocampi of mice, which was associated with ameliorated cognitive ability and object recognition memory during behavioral tests. Additionally, reverse transcription polymerase chain reaction (RT-PCR) revealed that the EGCG group showed significantly decreased expression levels of IL-1β mRNA compared to non-EGCG group, lending support to the mechanistic explanation that EGCG exerts anti-inflammatory effects via down-regulation of one or more inflammatory nuclear signaling pathways [36]. 

EGCG was shown to function as a more effective downstream inhibitor of inflammatory signaling compared to EGC and epicatechin (EC) [96]. Molecular docking analysis confirmed that while EGCG, EGC and EC were all able to occupy most of the TGFβ activated kinase 1 (TAK1) active site, a key inflammatory signaling node, only EGCG could also inhibit NFκB expression and p38 kinase, a key regulator in the inflammatory pathway. 

Human AC16 cardiomyocytes were pre-treated in vitro with EGCG 30 min before 4% cigarette smoke medium (CSM) was added to induce inflammation. EGCG treatment reduced CSM-induced inflammatory chemokine IL-8 production in the supernatant via the inhibition of ERK1/2, p38 MAPK and NFκB activation in AC16 cardiomyocytes [97]. It is interesting to note that down-regulating the ERK pathway in this study induced anti-inflammatory effects, while up-regulation of ERK has recently been identified as a primary target for antidepressants [71]. As a cellular pathway, ERK functions to communicate a signal from a receptor on the surface of the cell to the DNA in the nucleus, and as such can serve a multitude of functions, including the proliferation of inflammatory cytokines, or the potentiation of BDNF. Although currently in its early stages, the literature regarding the role of ERK signaling in depression pathology suggests that ERK more often represents a target for up-regulation in the context of BDNF-induced neurogenesis, rather than a target for down-regulation in the context of mediating inflammation. It is unclear at this time whether EGCG is capable of modulating multiple ERK pathways in different directions simultaneously, or what the net effect of EGCG-mediated ERK modulations is, considering the vast complexity of external signals that this pathway proliferates. 

Aside from EGCG and TF3, multiple other tea constituents have shown significant anti-inflammatory activity include gallic acid (GA), gallocatechin gallate (GCG), oolong tea extract, and L-theanine. GA reduced airway inflammation by decreasing the levels of IL-4, IL-5, IL-13 and IL-17 in nasal lavage fluid of mice with allergic rhinitis [102]. GCG, a tea catechin found in abundance in ready-to-drink (RTD) tea drinks, was found to inhibit LPS-induced expression of monocyte chemoattractant protein-1 (MCP-1) and IL-6 as effectively as EGCG in differentiated 3T3-L1 cells treated with LPS for 6 h [98]. It was found that GCG and EGCG both inhibited the LPS-induced phosphorylation of p65, and showed similar in vitro capacity to regulate NFκB activation. 

Oolong tea ethanol extract (OTEE) showed comparable anti-inflammatory activity to EGCG by reducing several inflammatory responses in LPS-induced murine macrophage cell line (RAW 264.7) [99]. This study found that, compared to a positive control group, OTEE and EGCG exhibited inhibitory activity toward levels of nitric oxide (NO), cyclooxygenase-2 (COX-2), TNF-α, IL-6 and IL-1β. Both OTEE and EGCG suppressed production of all of the aforementioned inflammatory cytokines and modulators; however, OTEE had higher inhibition activity than EGCG toward NO, COX-2, IL-6 and IL-1β, while the reverse was true for TNF-α. This lends support to the notion that chemical differences resulting from tea processing can cause metabolites to behave differently within certain specific pathways, yet produce similar net effects on larger target systems of antidepressant activity. 

One dose of topically delivered L-theanine was shown to reduce acute skin inflammation in a mouse model by inhibiting production of IL-1β, TNF-α and COX-2 [103]. An ovalbumin-induced murine model of asthma showed that L-theanine alleviated airway inflammation and dramatically attenuated extensive trafficking of inflammatory cells into bronchoalveolar lavage fluid (BALF) [104]. This study found L-theanine administration significantly decreased the production of MCP-1, IL-4, IL-5, IL-13, TNF-α, and interferon-gamma in BALF, exhibiting strong anti-inflammatory activity through suppression of the NFκB pathway. 

Human skin, in the form of cultured human epidermal keratinocytes, were treated with EGCG (3~10 μM) and exposed to airborne particulate matter PM10 (100 μg mL-1) in a study that showed anti-inflammatory effects of topical EGCG via suppression of PM10-induced TNF-α, IL-1β, IL-6, and IL-8 [100]. Another topical skin study found that anti-inflammation through the suppression of the NFκB /activator protein 1 (AP-1) pathway contributed to significantly improved acne in an eight-week randomized, split-face, human clinical trial using topical application of 1 and 5% EGCG [101]. 

Taken together, these results appear promising for the anti-inflammatory functions of several bioactive compounds found in tea, including TF3, EGCG, GCG, GA, teasaponin, oolong extract and L-theanine. However, the quantity and consistency of data remains insufficient to make conclusive claims on the viability of these compounds as anti-inflammatory agents in humans. However, even incremental improvements to chronic inflammatory states may be helpful in treating depression, as inflammation reverberates through to multiple downstream targets, including HPA activity and serotonergic neurotransmission. 

### 4.3. Restoration of Monoaminergic Systems

The monoamine hypothesis posits that depression results from dysregulated neurotransmission of one of more monoamines, including serotonin (5-HT), norepinephrine (NE) and dopamine (DA). This theory has been criticized as reductionist [106], and insufficient in considering the upstream factors that induce abnormal monoamine neurotransmission, such as chronic stress and inflammation. Despite the validity of these criticisms, the most common and effective current pharmacological approaches to treating MDD are monoamine reuptake inhibitors, such as selective serotonin reuptake inhibitors (SSRIs) or norepinephrine-dopamine reuptake inhibitors (NDRIs) [11], which function to increase circulating monoamine levels through the inhibition of their reuptake. These medications have been reported to be effective in preventing relapse of MDD in some cases, although data suggest that they are less effective as an isolated treatment than when used alongside other approaches, such as cognitive behavioral therapy [107]. While shortcomings of the monoamine theory are evident, it does not invalidate the notion that targeting monoaminergic systems is one sensible tactic in combating MDD.

As discussed in the previous section, inflammation significantly reduces circulating 5-HT levels, and individuals with MDD are likely to be found with dampened serotonergic systems. It was found that denervation of serotonergic projections reduced the generation of granule cells in the dentate gyrus of the hippocampi of rats, and these effects were reversed by stimulating 5-HT1A receptors or by boosting activity in 5-HT fibers, suggesting the close interconnection between serotonergic systems and hippocampal neuronal health [108] (Figure 3). Additionally, the mesolimbic pathway, consisting of dopaminergic neurons originating in the ventral tegmental area and projecting to the nucleus accumbens, amygdala and hippocampus, mediates motivation and the reward pathway. Numerous data have implicated altered dopaminergic neurotransmission within the mesolimbic pathway in the neuropathology of depression [109]. Norepinephrine (NE) is released together with cortisol from the adrenal gland following stimulation by ACTH. As with cortisol, NE can become chronically elevated under circumstances of constant perceived stress, driving the release of cytokines, which can exert reciprocal effects on the HPA axis [23]. 

Finally, there exists interconnectivity among 5-HT, DA and NE, wherein any action on one system has the potential to reverberate through to the other systems [110]. For example, DA can inhibit the release of NE from the locus ceruleus. NE can have excitatory and inhibitory effects on DA release in the ventral tegmental area through stimulation of α-1 and α-2 receptors, respectively [23]. Likewise, both NE and DA can act on α-1 and D-2 receptors, respectively, to increase 5-HT release from the dorsal raphe nucleus. As more recent data show, these monoaminergic systems show the potential to be modulated by microbiota species in the intestines, characterizing a relationship known as the gut–brain axis. Taken together, these data indicate that restoration of monoaminergic systems represents an important and interconnected node within the larger network of MDD pathology. Reviewed here are studies that measured how compounds in tea interact directly with monoaminergic systems, and indirectly through their modulations of GM, including mechanistic explanations for the observed effects (summarized in Table 3).

A recent study measured the effect of L-theanine (2 mg/kg) on 5-HT, NE and DA levels in an animal model of depression [111]. Measurements were taken in the limbic-cortical-striatal-pallidal-thalamic (LCSPT) circuit, which included the prefrontal cortex (PFC), nucleus accumbens (NAC), striatum (ST), amygdala, and hippocampus (HIP). The LCSPT circuit was chosen for this study as multiple previous studies showed that depressive symptoms can derive from dysfunctional monoamine neurotransmission relating to LCSPT circuitry [124,125]. L-theanine ameliorated depressive-like behaviors in a chronic unpredictable mild stress (CUMS) rat model, which is a validated rodent model widely used for analyzing mechanisms underlying depression [126]. In the PFC, NAC, and HIP, L-theanine administration significantly increased the levels of 5-HT, NE, and DA. In the ST, the levels of 5-HT and DA were significantly increased following theanine supplementation. In the HIP, only DA levels significantly changed after theanine treatment. Despite the notable results, the authors were unable to identify the detailed signaling pathways responsible for the observed theanine-mediated improvements in monoamine neurotransmission [111]. 

Zhu et al. [112] found that, in an Alzheimer’s Disease (AD) mouse model, supplementation with L-theanine promoted hippocampal dopamine and noradrenaline release, and that these effects were blocked by antagonists of the dopamine D1/5 receptor, suggesting that theanine improved hippocampal monoamine transmission via dopamine D1/5 receptor-PKA pathway activation. This proposed mechanism could potentially explain why DA was found to be the only monoamine upregulated in the hippocampus following L-theanine supplementation in the aforementioned study by Shen et al. [111].

Rats that had been administered green tea for five weeks prior to a single restrained stress were found to be more capable of coping with the stressful conditions/stimuli than rats who had not been administered green tea prior to the stressful condition. Stress-induced anxiogenic effects were attenuated and tone of 5-HT was normalized in green tea-treated rats, suggesting that altered 5-HT metabolism allowed rats to cope with stress more effectively [113].

Four milligrams per day of epicatechin in the water of C57BL/6 mice reduced anxiety in the open field (OF) and elevated plus maze (EPM), which was reflected by downregulated cortical monoamine oxidase (MAO)-A levels, but not MAO-B, and increased BDNF levels in the hippocampus but not the cortex. Additionally, EC significantly increased hippocampal and cortical levels of tyrosine hydroxylase (TH), a critical enzyme for monoamine synthesis [114]. EC ingestion did not facilitate neurogenesis in the dentate gyrus, suggesting a non-neurogenic mechanism. Interestingly, this study found that BDNF up-regulation resulted from increased levels of protein kinase B (Akt), rather than ERK/CREB signaling. The measured anxiolytic effect resulting from EC intake was thus interpreted to have resulted from concurrent modulation of complimentary neurotrophic and monoaminergic signaling pathways. 

Using the CUMS-induced depression model in mice for five consecutive weeks, tea polyphenols (TP) were administered at a daily dose of 25 mg/kg or 50 mg/kg by gavage for three consecutive weeks starting from the third week [115]. The results show that CUMS-induced depression significantly decreased 5-HT and NE in the hippocampus and PFC, and TP administration effectively reversed the alterations in concentrations of 5-HT and NE, in turn exhibiting significant anti-depressant effects in mice with CUMS-induced depression. Zhao et al. [35] also found that administration of either EGCG or GTP were associated with partial restoration of normal DA and 5-HT levels in mice following a four-week restraint stress-induced neural injury. Although reduction in stress-induced neural injuries were attributed to restoration of the ERK1/2 signaling pathway, it is unclear which mechanism mediated TP-induced improvements to monoaminergic systems in either of these studies.

Black tea TF exerted an anxiolytic effect in mice by increasing DA turnover in the frontal cortex, as measured by increased levels of 3,4-dihydroxyphenylacetic acid (DOPAC) and the ratios of DOPAC/DA. These results suggest that TF may be able to induce anxiolytic effects via activation of the dopaminergic system in the frontal cortex [120]. As the following study suggests, TF-induced improvements to DA neurotransmission may be mediated through reduction in free radical damage to these systems, and corresponding improvements in DA transporter function.

A mouse model measured the protective effects of black tea TF on monoamine transporters and behavioral abnormalities in 1-methyl-4-phenyl-1,2,3,6-tetrahydropyridine (MPTP)-induced neurodegeneration [121]. MPTP-injection has been shown to enhance lipid peroxidation in striatum via enzymatic conversion of MPTP to MPP+, inducing the generation of free radicals [127]. Administration of MPTP (30 mg/kg bw for four consecutive days) led to reduced behavior patterns (open field, rotarod and hang test), and reduced expressions of dopamine transporter (DAT) and vesicular monoamine transporter 2 (VMAT2). TF supplementation included 10 mg/kg orally for three days, followed by the same dose taken 1 h before intra peritoneal injection of MPTP for four days, totaling a seven-day experimental period. Pre-treatment with TF significantly reduced oxidative stress and preserved striatal DA levels by preventing the degeneration of dopaminergic neurons by MPTP, which resulted in improved motor behavior and improved expression of DAT and VMAT2 in striatum and substantia nigra. These data may highlight the importance of free radical scavenging in mediating the protective effects of tea polyphenols within multiple mechanistic pathways simultaneously. In other words, the anti-oxidant capacities of tea polyphenols may represent at least some portion of the net benefit of tea consumption conferred to both inflammation reduction and restoration of monoaminergic systems. 

#### 4.3.1. Monoamine Oxidase Inhibition

Although circulating monoamine levels have been found to improve following tea intake, the data implicating the role of monoamine oxidase (MAO) inhibition as a pathway for these modulations have been inconsistent. Two isoenzymes of MAO have been found in humans, including MAO-A, mainly responsible for the breakdown of 5-HT and NE, and MAO-B, which has relative affinity for DA metabolism [128]. Inhibition of this class of enzymes by various drugs and plant compounds arrests the breakdown of neurotransmitters, increases their concentration in the brain and can bring about antidepressant effects in individuals with MDD [129,130]. 

One in vitro study showed that none of the common green tea catechins, i.e., EC, EGC, EGC, and EGCG, possess inhibitory effects on MAO-A activity in mouse brain mitochondria [131]. However, an in vivo mouse model showed that four weeks of oral EC supplementation was able to decrease MAO-A expression, but no effect was found on MAO-B [114]. However, MAO-B has been shown to be inhibited to various degrees by several types of black tea and green tea in vitro [132], and EGCG has been shown to inhibit MAO-B in brains of rats [133]. Despite the contradictory data, there is a general consensus among the aforementioned studies, and others [134,135], that tea polyphenols can at best confer modest decreases in MAO activity, however this marginal effect is likely insufficient to consider it a primary mechanism driving the observed attenuations to monoaminergic systems following tea intake. However, more research is needed to examine the impact of various tea compounds and combinations of tea compounds on the in vivo attenuation of MAOs, particularly MAO-A, due to stronger association of that isoenzyme with depression and other mood disorders [130]. 

#### 4.3.2. Modulating Monoamines via the Gut–Brain Axis

The gut–brain axis is a bi-directional communication network connecting the central nervous system (CNS), to the enteric nervous system (ENS) [136]. A complex reflexive network of efferent fibers projecting into the gastrointestinal (GI) tract and afferent fibers that project into a number of interconnected regions of the CNS facilitate communication within the axis [137]. Within this network, changes in gut microbiota can trigger changes in brain chemistry, including neurotransmitter levels [138]. Absence and/or modification of the gut microbiota (GM) in mice affects the HPA axis response to stress [139,140] and anxiety behavior [141,142]. For example, germ free (GF) mice, i.e. mice that were born and bred in entirely germ-free environments, had significantly lower 5-HT levels and overall 5-HT receptor expression in the amygdala and hippocampus than mice that were raised in a probiotic environment [143,144,145]. Tea polyphenols and aromatic tea compounds have been shown to be capable of exerting profound effects in the gut by inducing growth of GM strains, or “psychobiotics”, that modulate the gut–brain axis in beneficial ways to mental health and depression [146,147,148]. 

Tea polyphenol intake (400 mg/volunteer, three times per day, for four weeks) increased *Bifidobacterium* spp. (the acid forming bacteria) in total counts in a human trial [149]. It should be noted that this study is from 1992, and recent estimates place the tolerable upper-intake level of tea polyphenols around 400 mg per day [150]. More recently, 4% green tea powder supplementation for 22 weeks in HFD mice significantly increased *Lactobacillus* species, in both number and diversity [151]. After incubation with human fecal samples, (+)-Catechin (150 mg/L exposure) in vitro lead to increases in both *Lactobacillus* spp. and *Bifidobacterium* spp., although increases only achieved significance for the latter [122]. 

*Bifidobacterium* spp. have been associated with beneficial psychobiotic effects, including *Bifidobacterium longum* 1714, shown to have effectively attenuated cortisol output, daily reported stress levels, and subjective anxiety in response to the socially evaluated cold pressor test in healthy human volunteers (*n* = 22) [152]. Another *Bifidobacterium* strain, *Bifidobacterium infantis*, reversed HPA axis hyperactivity in GF mice as measured by plasma ACTH and corticosterone in response to restraint stress [139]. It was found that supplementation with O-Methylated EGCG (EGCG3”Me) isolated from oolong tea leads to significant increases in *Bifidobacterium* compared to mice not treated with EGCG3”Me, suggesting a modulatory effect of EGCG3”Me on *Bifidobacterium* spp [123]. An earlier study by the same authors showed that EGCG, GCG, and EGCG3”Me isolated from Chinese oolong tea promoted the growth of *Bifidobacterium* spp. and *Lactobacillus/Enterococcus* groups when incubated with human intestinal microbiota in vitro [118]. A preparation of 100 mg of mixed tea catechins (approximate to a cup of green tea) three times daily for three weeks induced a significant increase in *Lactobacillus* species in broiler chickens [153].

*Enterococcus* species have been shown to generate 5-HT [142], while GABA can be secreted by both *Lactobacillus* and *Bifidobacterium* strains [154]. Bravo et al. [155] demonstrated that chronic administration of probiotic *Lactobacillus rhamnosus* improved GABA receptor expression in cortical regions and the hippocampus, which reduced stress-induced corticosterone and depression behavior. Importantly when the vagus nerve was removed from mice, the neurochemical and behavioral effects were not found, indicating the vagus nerve as a major modulatory communication pathway for psychobiotics. One recent study found that a *Lactobacillus* strain, *Lactobacillus plantarum*, was able to reverse reduced DA and 5-HT levels in depressed mice and increase levels of BDNF [156]. These data suggest that tea polyphenol-induced modulations to *Lactobacillus*, *Bifidobacterium*, and *Enterococcus* species may be able to explain some of the improvements to monoaminergic systems, HPA hyperactivity, and BDNF that have been observed following tea intake.

A mechanism by which tea polyphenols may be functioning to increase the levels of probiotic strains such as *Bifidobacterium* or *Lactobacillus* may be by improving their adhesion to intestinal mucosa, which is considered as a prerequisite for colonization and an important characteristic when measuring the ability of probiotic strains to confer health benefits. Bustos et al. [119] investigated the ability of several tea polyphenols to improve adhesion of *Lactobacillus* strains to human epithelial intestinal cell lines. Both EGCG and EGC significantly increased adhesion of different *Lactobacillus* strains to intestinal cells, however some tea polyphenols did not, depending on *Lactobacillus* strains, and the epithelial cell lines tested in the study.

Tea polyphenols have shown variable effects on *Bacteroides*, another potentially psychobiotic species whose abundance was shown to negatively correlate with brain signatures associated with depression [157]. EGCG, GCG and EGCG3”Me cultured with human intestinal GM in vitro were found to inhibit the growth of *Bacteroides* [118]. Another study found that EGCG3”Me inhibited growth of *Bacteroides* in HFD mice between Weeks 0 and Week 3 of the experiment, but *Bacteroides* began to proliferate starting from Week 4 of EGCG3”Me supplementation [123]. It is still unclear what the overall effect of polyphenol intake is on *Bacteroides* species, and equally important, what the overall effect of *Bacteroides* species is on human health [158]. *Bacteroides* have been shown to be elevated among individuals with obesity [159], and reduced among individuals that underwent six weeks of exercise training [160], suggesting that *Bacteroides* can correlate with measures of poor human health, despite their association low brain signatures of depression. The inconsistency in these data may be because *Bacteroides* are typically more abundant among people with long-term diets rich in animal protein and saturated fats [161], and less abundant among people with plant carbohydrate-based food diets, which remains a poorly understood and highly-contested topic. Taken together, multiple factors need more careful analysis regarding the tea-mediated effects of GM on depression, including which species of GM can serve as realistic modulatory targets for tea polyphenols, and the mechanisms by which those targeted GM species can attenuate MDD pathology.

#### 4.3.3. Generation of Short-Chain Fatty Acids

Short-chain fatty acids (SCFAs) are another factor in the tea–gut–brain axis, as they are products of microbial fermentation of undigested carbohydrates in the GI tract, and can increase as a result of tea intake [162]. The three principal SCFAs (butyrate, acetate and propionate) can cross the blood brain barrier [163,164] where they have been shown to alleviate psychosocial stress-induced alterations in behavior, and exhibit antidepressant and anxiolytic effects in mice [165]. One proposed mechanism of procognitive and beneficial behavioral effects of butyrate has been suggested to be through its potent inhibition of histone deacetylases (HDACs), promoting histone acetylation and stimulation of gene expression in host cells [166].

In a human trial, high doses of tea polyphenol intake (400 mg/volunteer, three times per day, for four weeks) significantly increased levels of acetic and propionic acids [149]. A study researching the in vivo effects of tea polyphenol supplementation in HFD mice reported that butyric acid more than doubled after black tea polyphenol (BTP) supplementation compared to the control group [117]. Black tea in particular, more so than green tea, has been shown to be highly effective in generating colonic SCFA due to the high molecular weight of black tea polyphenols, which generally remain undigested until they reach the colon and interact with GM [162]. Additionally, Henning et al. [117] showed that SCFA generation by both green tea and black tea polyphenols induced a significant increase in hepatic 5′adenosylmonophosphate-activated protein kinase (AMPK), by 70% and 289% for green and black tea, respectively. We previously characterized the chain of mechanisms by which oxidized polyphenols may be more effective promoters of AMPK than unoxidized polyphenols, through relatively strong inhibition of carbohydrate digestive enzymes, α-glucosidase and α-amylase, which creates more substrate for SCFA generating interactions in the colon, leading to greater AMPK activation [162]. AMPK activation was recently shown to ameliorate depressive-like behaviors in mice by elevating hippocampal neurogenesis potentially via PKC zeta/NF-κB/BDNF/TrkB/CREB signaling in neurons [167].

In a HFD mouse model, EGCG3”Me significantly increased butyrate-producing bacteria such as *Eubacterium*, *Ruminococcaceae* (including *Faecalibacterium*) and *Roseburia*, and the majority of acetate-producing bacteria such as *Coprococcus* and *Bifidobacterium* [123]. Zhang et al. [118] showed that EGCG, GCG and EGCG3”Me were able to induce higher concentrations of SCFA compared to control within a human fecal incubation experiment. Taken together, SCFA represents an interesting piece of the gut–brain axis that may have relevance to the antidepressant activity in tea. However human trials are necessary given the vast differences between human and mouse GM, and great inter-individual variability among human GM.

### 4.4. Restoring Neurogenesis and Neuroplasticity

As mentioned above, chronic stress and inflammation lead to hypercortisolemia, or chronic elevated cortisol levels with neurotoxic effects, leading to reduced neurogenesis, as seen in neuroimaging studies showing reduced volume in the prefrontal cortex (PFC), especially in the subgenual PFC, basal ganglia and hippocampus of patients with MDD [168,169]. Additionally, natural age-related stress in the brain [170,171,172], or environmentally imposed stressors such as air pollution [173] can cause oxidative damage to neurons [174] and reduce their functionality, a process that has been linked to depression in meta-analyses [175]. The regions of the brain including the amygdala, hippocampus, basal ganglia, and nucleus accumbens are responsible for mediating raw, unprocessed emotion, while areas of the PFC mediate cognitive processing of emotions [176]. With reduced volume and interconnectivity among these critical emotion-processing centers, individuals become less capable of executive function and mood regulation, a proposed pathophysiology of depression [177]. Meta-analyses of human neuroimaging studies showed that patients with MDD had increased activation of the interior cingulate and amygdala and decreased activation of striatum and dorsolateral PFC, which presented as hyperactivation for negative stimuli and hypoactivation for positive stimuli [178]. The impeded ability to process emotion properly causes an individual to feel more overwhelmed, respond more reactively to external stimuli, all of which further fuel the cycle of chronic stress/inflammation. For these reasons, fighting the stress-induced reduction in neurogenesis has been said to be critical to the efficacy of antidepressant treatment [179].

Brain-derived neurotrophic factor (BDNF) is a neurotrophin that promotes the survival of existing neurons and encourages the growth and differentiation of new neurons and synapses [180]. It was found that serum levels of BDNF are reduced in patients diagnosed with MDD [181] and that depressive behavior could be induced in rats following a knock-out experiment of BDNF in the dorsal dentate gyrus [182]. BDNF has been shown to protect against stress-induced neuronal damage [183] and affect hippocampal neurogenesis [184], which has led some to suggest that BDNF may be an important agent for therapeutic recovery from MDD [185,186,187]. Conceptually, the role of BDNF and neurogenesis in treating MDD is to help the brain return to a state of juvenile-like plasticity by encouraging the growth of new neurons and synapses that can be utilized to establish new pathways for emotional processing which mitigate stress and encourage healthy cognitive behavior. Here, we review studies that have measured how compounds in tea have been observed to interact with neurons to affect BDNF or neuroplasticity, including the proposed mechanisms for those effects (summarized in Table 4).

Eight weeks of high-fat diet (HFD) in C57BL/6J mice followed by treatment with oral teasaponin (0.5%) mixed in HFD for another six weeks effectively prevented HFD-induced recognition memory impairment, and improved neuroinflammation and BDNF deficits in the hippocampus, in addition to attenuating gut microbiota alterations [105].

Compared to a control group, newborn rats fed water containing 0.3% theanine for six weeks showed increased exploratory activity in the open-field test with enhanced object recognition memory, and significantly increased BDNF levels accompanied by significantly increased number of BrdU-, Ki67-, and doublecortin (DCX)-labeled cells in the granule cell layer, suggesting that theanine administration facilitated neurogenesis and improved plasticity in the developing hippocampus [188]. In corroboration with these results, DCX-positive neurons showed more elaborate dendritic trees, accompanied by significantly increased hippocampal neurogenesis following 2.5 mg/kg EGCG treatment in mice. BDNF was not measured in this study [189].

Three-month-old C57BL/6J mice were fed a high-fat/high-fructose diet (HFFD), which induced peripheral inflammation and neuronal loss/damage [95]. EGCG supplementation (2 g/L), in drinking water) attenuated HFFD-induced neuronal damage and reduced cognitive disorder by significantly up-regulating the ERK/CREB/BDNF signaling pathway. This study may provide compelling evidence that EGCG can restore HFFD-induced hippocampal changes and reverse neuronal damage via significant upregulation of the ERK/CREB/BDNF signaling pathway [95].

Gundimeda et al. [190] showed that in PC12 cells treated in vitro with even submicromolar concentrations of EGCG (<0.5 μM) potentiated the neuritogenic ability of a low concentration (2 ng/mL) of BDNF. Furthermore, it was observed that, while EGC and EC were unable to significantly potentiate the neuritogenic ability of BDNF individually, both catechins enhanced the BDNF potentiating capacity action of EGCG.

A recent study used adrenal phaeochromocytoma PC12 cell line to study the neuronal damage induced by corticosterone in vitro, which is a common in vitro experimental model of depression-induced neuronal damage [191,199]. Results showed that exposure to high concentrations of corticosterone induced cytotoxicity and downregulated the Sonic hedgehog pathway (Shh) in PC12 cells, and that these effects were attenuated by administration of EGCG. Interestingly, this study found the optimal concentration of EGCG for attenuating the corticosterone-induced decreases in PC12 cell viability to be 20 μM, however Pervin et al. [192] found the optimal concentration of EGCG for SH-SY5Y cell growth in vitro to be 0.05 μM, with diminishing effects at higher concentrations. Similarly, Gundimeda et al. [190] showed that EGCG was able to induce BDNF-mediated neuritogenic growth at submicromolar concentrations (<0.5 μM). Inconsistencies in data may be due to the use of PC12 rodent cell lines in one study, and human-derived SH-SY5Y cell lines in the other. Additionally, these studies examined the activation of different pathways, one measuring the Shh pathway and another measuring a BDNF pathway. Thus, these inconsistencies highlight that that optimally efficacious level of tea intake may vary considerably depending on which mechanistic pathway, in addition to which animal species, the tea compounds are functioning within.

Interestingly, we found one study that showed negative results for the ability of EGCG to induce neurogenesis in mice [200]. This study used a dosage of EGCG that was an order of magnitude higher than the dosages used in the other experiments reviewed here (~250 mg/kg/day), which in a 70 kg human translates to a dose of 1422 mg EGCG [201], or over double the estimated toxic dose (600 mg) and four-fold standard estimates for acceptable daily intake (300–400 mg/day) [150]. The finding that this large dosage was ineffective at inducing neurogenesis in mice lends support to the research by Pervin et al. [192,202] who measured a diminishing ability of EGCG to induce cell growth at high doses.

To measure the effects of non-EGCG tea catechins on age-related changes in rat hippocampal formation (HF) and behavioral alterations, Rodrigues et al. [197] prepared a green tea extract (GTE) rich in catechins and poor in EGCG. GTE-treated mice presented plastic changes in dendritic arborizations of dentate granule cells accompanied by improved spatial learning in the Morris water maze, demonstrating that non-EGCG catechins protected HF from aging-related declines in structural formation with repercussions on behavioral performance.

Unno et al. [196] showed the primary EGC metabolite, EGC-M5, and several of its conjugated forms, led to improvements in human neuroblastoma SH-SY5Y cells, where neurite length was significantly prolonged by EGC-M5, and the number of neurites was increased significantly by all metabolites examined [196]. This study also examined in vitro BBB permeability of EGCG, finding that EGCG and the combination of EGCG’s hydrolysis products, EGC and gallic acid (GA), can pass through BBB and effectively induce neurite growth. The combination of EGC and GA worked more efficiently than either one alone. This finding suggests that ingestion of EGCG can provide neurogenic benefits even after undergoing in vivo hydrolysis to EGC and GA.

In corroboration with the reported effects of GA, an in vivo mouse model found that GA ameliorated trimethyltin chloride (TMT)-induced anxiety and depression in behavioral tests, and that cell densities in the CA1, CA2, CA3 and DG hippocampal subdivisions from GA-treated rats were higher than in saline treated rats [198]. Several additional in vivo mouse models have found EGCG or the combined effect of green tea catechins to significantly upregulate BDNF expression, resulting in neurogenic effects in hippocampal neurons [193,194,195,203].

Taken together, these data suggest that even small doses of teasaponin, L-theanine, gallic acid, and both EGCG and non-EGCG tea catechins can exert neurogenic effects, at least in mice and in vitro models. An important finding is that that there appears to be an optimal dosage for EGCG-induced neurite outgrowth, and excessive intake of EGCG may hinder the ability of EGCG to promote neurogenesis. These findings are encouraging for the potential viability of neurogenic mechanisms to mediate antidepressant effects of tea consumption. However, more data are needed to examine the mechanisms involved and dosages necessary to optimize tea-induced neurogenesis.

### 4.5. Mood Enhancement

It may be that, for some people, the very activities of preparing and consuming tea bring about mood-enhancing effects that are separate from and additional to the physiological effects within the four nodal systems discussed above. For those who derive pleasure from the taste, aroma, aesthetics, or the moment of time taken away from a busy routine to stop and consume tea, there may be improvements to affective state that confer anti-depressive effects by their own right. It has been shown that positive affective states build enduring personal and cognitive resources [204] and higher levels of creativity compared to neutral mood states in meta-data from mood-creativity research [205]. Furthermore, mood may impact not only cognitive control processes, but also attention, perception, and memory [206,207,208]. Likewise, neuroimaging data suggest that a reciprocal relationship may exist, in which mood affects cognition, but cognition also affects mood [209], and that components of this neural network are predictive of response efficacy to MDD treatment [210].

In a clinical trial, greater reported happiness throughout the day was associated with lower salivary cortisol on both working and nonworking days, reduced fibrinogen stress responses, and lower ambulatory heart rate in men [211]. Positive mood induction significantly decreased psychological distress and emotional processing deficits compared with a neutral mood induction condition in a trial studying depressed stroke subjects (n = 30) and rheumatic/orthopedic controls (*n* = 30 [212]. These data suggest that small daily inductions of positive affect, creativity and overall cognition may effectively accumulate over time and contribute to long-term protective effects against MDD. Here, we review studies that have measured effects on positive affect resulting from tea consumption, including mechanistic explanations for the observed effects.

At the meta-data level, tea intake has not been strongly associated with improved affective state, with the exception of improved alertness [213,214], likely resulting from synergistic effects of caffeine and L-theanine [215]. One randomized, placebo-controlled, single-blind study (n = 23) conducted in four sessions found that 4 g of matcha tea powder induced no significant changes in mood, but did induce significant improvements in attention tasks and psychomotor speed in response to stimuli [216]. In two studies measuring the effect of EGCG on affective state, one reported no mood effect in response to 270 mg EGCG tablets [217], and another found 300 mg EGCG supplementation to significantly increase self-rated calmness (*p* = 0.04) and reduced stress (*p* = 0.017) measured as change-from-baseline [218].

Einother et al. [219] showed that a regular tea consumer preparing and drinking a single cup of tea experienced significant feelings of pleasure. However, Einother et al. [220] later measured the pleasurable effects of the same quantity of tea, and found no effect on pleasure. Notably, the differences between the first and second studies were twofold; participants in the first study were regular tea drinkers, and also they were involved in the preparation of the tea they consumed. Participants determined their own steep-time, addition of milk/sugar, stirring, etc. These differences in experimental setup are significant, and results are concomitant with data showing that simple preparatory tasks can heighten product involvement and thereby pleasure experienced [221,222]. Being a regular tea consumer was likely an influential factor, considering the importance of anticipatory (appetitive) positive affect state [223], and the neural correlation of desire, wanting, liking, etc. with pleasurable effect of the wanted [224]. While these data suggest that the act of tea consumption will not confer mood-enhancing benefits to everyone, they do suggest that, for individuals who have rituals or habits surrounding tea consumption, this activity can be mood-enhancing, and perhaps through that mechanism, anti-depressive.

There are numerous factors modulating the potential change to affective state due to tea consumption. The expectation itself, resulting from having ingested caffeine has been shown to improve mood as well as performance [225]. Physical warmth of holding a warm beverage has been shown to affect social information processing, leading to increased perceived warmness of another person [226] and perceived emotional connection [227], although the connection between physical and emotional warmth has been disputed [228]. Compounds with bitter taste were found to be negatively correlated with mood scores, and there may be potential asymmetry in the effects of bitter and sweet taste modalities on mood [229]. However, the effect size in the mentioned study above was dependent on participants’ perception of the compound as bitter, which is a sensitivity with great inter-individual variability based on variations in bitter taste receptor subtypes [230,231,232], which are differentially expressed throughout the population [233,234] based on multiple factors including age and genetics [235].

Cultural habits may also impact the mood effects conferred by tea consumption. Firstly, Western nations have historically built habits around adding milk and sugar to tea [236], which has been found to mitigate many of the beneficial health effects of tea polyphenols [237,238], likely due to a condensation reaction between the hydroxyl group of the phenolic compounds and the hydroxyl group of the sucrose molecule leading to the formation of a glycoside. Additionally, due to the rich content of bitter polyphenol compounds in tea, and the high-fructose nature of the “Western diet”, it is possible that tea may be more easily perceived as bitter to Western drinkers, or other populations that tend to consume sweet foods. However, despite the Western diet, tea consumption remains at strong or increasing popularity in Western nations such as Germany and the United States [239,240].

Taken together, these data suggest that mood-enhancement as an additional mechanism for mediating antidepressant activity may be viable for some individuals, but less viable for others, depending on a large number of variable and subjective factors. It may be an interesting topic for future research to design mood-enhancing, mindfulness-based, or otherwise therapeutic rituals surrounding daily tea consumption, which could function to confer benefits to affective state on top of the number of neurobiological benefits we have discussed.

### 4.6. Determining Bioavailability, Pharmacokinetics and Pharmacodynamics

In 2010, Del Rio et al. [241] discovered that the overall bioavailability of green tea flavan-3-ols was an order of magnitude higher than previous estimations ranging from 2% to 8.1% [242,243]. Del Rio found bioavailability averaged 39% in humans after considering polyphenol metabolites generated from interactions with colonic microbiota [241]. Examining the interactions between polyphenols and the GM is critical to understanding the bioavailability of these compounds in the body, since 90–95% of dietary polyphenols have been shown to accumulate and break down in the large intestine [244]. Since discovering the importance of GM to polyphenol bioavailability, more research has emerged which seeks to measure the mechanisms by which GM cleave flavonoid conjugates and glycosides, thereby modulating their health effects [245,246,247]. Shang et al. [248] showed in rats that a single oral administration of EC (350 mg/kg) underwent in vivo metabolic reactions, including methylation, dehydration, hydrogenation, glucosylation, sulfonation, glucuronidation and ring-cleavage, resulting in 67 metabolites found in urine [248].

Recently, the first in vivo human experiment was conducted using a radiolabeled epicatechin isotope (2-C-14) (EC14) to track flavonoid bioavailability in the human body [249]. Eight male volunteers each ingested a drink containing 207 μM radiolabeled EC14, after which blood and urine samples were attained over a 48-h period. It was found that EC14 was absorbed in the small intestine as 12 structurally-related epicatechin metabolites (SREMs), attaining sub-μM/L peak plasma concentrations (C-max) around 1 h after ingestion. SREMs were excreted in urine over a 24 h period in amounts corresponding to 20% of EC14 intake. On reaching the colon, EC14 underwent GM-mediated conversions yielding the 5C-ring fission metabolites (5C-RFMs), which appear in plasma as phase II metabolites with a C-max of 5.8 h after intake, and were excreted in quantities equivalent to 42% of the ingested EC14. Other catabolites excreted in 0–24 h urine amounted to 28% of intake. Overall EC was found to be highly bioavailable, with urinary excretion indicating that 95% is absorbed and passes through the circulatory systems as a diversity of phase II metabolites [249]. This study was the first to track flavonoid metabolism in humans with such a high degree of specificity. However, no study has yet tracked these metabolites for in vivo penetrability of a human BBB. For now, only animal models can provide insight into the bioavailability of flavonoid metabolites in the brain.

Wang et al. [250] fed rats with 17 mg/kg of a grape-derived polyphenolic preparation (GP) comprising proanthocyanidin, catechin and EC to test the effects of their metabolites on neurological disorders [250]. Following acute intake, low, unquantifiable amounts of EC metabolites were detected in rat brains. However, following 10 days of chronic supplementation, plasma metabolite levels increased substantially, and trace amounts of unmetabolized EC and a mixture of EC metabolites were found in the brain at a concentration of 484 nM, and potentially could have been higher if ring fission metabolites, which were not analyzed, were also present. Additionally, this study reported that a biosynthetic proanthocyanidin EC metabolite identified in the brain, 3′-O-methyl-epicatechin-5-O-beta-glucuronide (3′-O-Me-EC-Gluc), promoted basal synaptic transmission at physiologically relevant levels in hippocampus slices through mechanisms associated with CREB/BDNF signaling [250]. If these data were taken together with research by Unno et al. [196] showing the ability of the EGC metabolite, EGC-M5, and several of its conjugated forms, to cross a human BBB model, and data from Pervin et al. [192] showing an ideal concentration of 50 nM of EGCG for SH-SY5Y cell proliferation, it would appear at least feasible to hypothesize that in human conjugated polyphenol metabolites can cross the BBB, and accumulate in human brains at levels of physiological relevance to mitigating MDD neuropathology [192,202]. However, a great deal of research is still needed to understand the bioavailability and brain permeability of various tea polyphenols and their metabolites.

## 5. Discussion

While it appears promising that tea consumption can confer mild anti-depressive effects through a number of mechanisms, it remains unclear as to which specific types of tea (i.e., green, white, black, oolong, yellow, and dark) produce maximal net effects on risk reduction of depression. We hypothesize that consuming multiple tea types regularly would confer greater effects than consuming equivalent amounts of only one tea type. For instance, green tea possesses a higher anti-oxidant capacity relative to other teas due to its high total phenolic content [251], however the high molecular weight oxidized tea polyphenols, such as those in oolong and black teas, have shown to be relatively effective promoters of SCFA generation and AMPK activation [162].

Wang et al. [252] tracked the metabolite profiles of an identical batch of fresh leaves that was processed into each of the six major tea types, and the results may provide insight as to the relative capacities of certain tea types to confer anti-depressive effects through different mechanisms [252]. This study found significant differences in the profiles of certain metabolites with considerable relevance to the antidepressant mechanisms discussed in this review. For example, gallic acid (GA) increased in white tea, black tea and dark tea, likely due to “crack reaction” products resulting from fermentation and post-fermentation processing [253,254]. Wang et al. [252] found that white tea processing lead to two-fold increases in tryptophan and GABA, and these amino acids were found mostly to decrease, or only increase marginally, in other tea types. Tryptophan as the serotonin precursor, and high-GABA content tea, shown to alleviate depression and stress in mice [68,255], both may be relevant to added anti-depressive effects conferred by white tea. Following black tea processing, there was a 12.73-fold decrease in EGCG, coinciding with a 30.74-fold increase in theaflavin digallate, which may carry considerable implications for reasons already discussed. An interesting area of future research may be examining the isolated effects of certain tea types within a mechanistic system, and then comparing the synergistic effects of two or more tea types within that same system. We may find that diversity in tea is highly beneficial, just as diversity in other forms of dietary intake.

## 6. Conclusions

This article critically examines the biochemical and neurobiological effects of tea consumption vis-à-vis our current understanding of the mechanisms purported to underlie depression. Prior research has documented an inverse relationship between tea consumption and depression. Based on this review, it appears that many biochemical and neurobiological effects of tea mimic anti-depressant processes within our current depression pathology framework.

In summary, there appear to be multiple pathways affected by multiple constituents in tea that can collectively lead to antidepressant effects in tea drinkers. Notable among these pathways is the ERK/CREB/BDNF signaling pathway, which has been shown to be up-regulated by a number of compounds in tea including teasaponin, L-theanine, EGCG, EC and combinations of green tea catechins and their metabolites. Other tea-mediated mechanisms include increased SCFA/AMPK signaling, notably effective in oxidized tea polyphenols, and improved generation of monoamine and BDNF-up-regulating “psychobiotic” bacterial strains in the GM, such as *Lactobacillus* and *Bifidobacterium* species. Additionally, reduced inflammation, improved monoaminergic systems, and reduction of stress response via normalized HPA axis activity all represent major nodes of tea-mediated antidepressant activity. However, at this time, there are not enough data to specify which mechanisms are responsible for producing the largest net induction of antidepressant response, which might serve as an interesting topic for future research. This review is the first to use modern integrated theories of depression neurobiology to comprehensively map out the mechanisms through which tea may be working to reduce risk of depression. In conclusion, the current data suggest that daily consumption of moderate amounts of tea may offer significant potential benefit in the risk reduction of depression.

## Figures and Tables

**Figure 1 nutrients-11-01361-f001:**
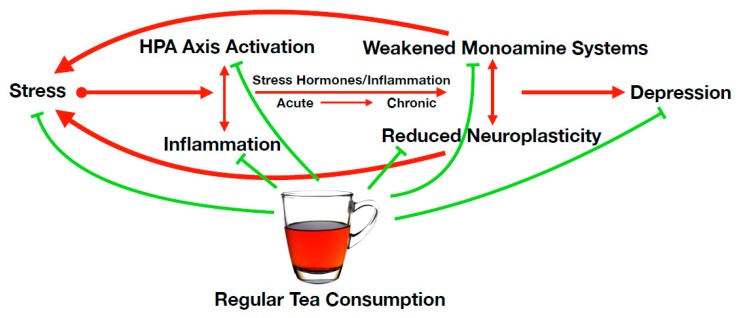
The effects of regular tea intake within a unified theory of depression pathology. External stressors induce an HPA-mediated stress response and inflammation. If acute stress and inflammation become a chronic, persisting physiological state, there can be detrimental effects on neuronal health and monoaminergic systems. Compromised cognitive emotional processing resulting from cumulative neuro-pathologies inhibits the ability to cope with future external stressors, re-feeding the state of chronic stress/inflammation. Green lines represent attenuating effect, while red lines represent exacerbating effect. HPA, hypothalamic–adrenal-pituitary.

**Figure 2 nutrients-11-01361-f002:**
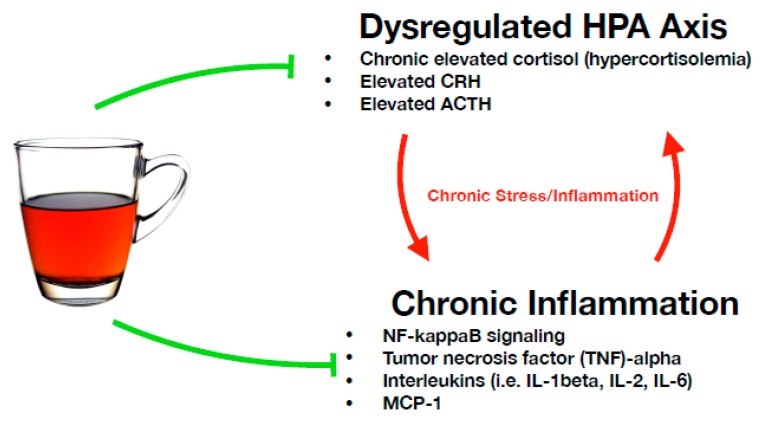
The positive feedback loop of chronic stress and inflammation. Dysregulated HPA axis and chronic inflammation represent two mechanistic nodes within an interrelated system of depression pathology. Bullet points represent signaling pathways and pathological targets for tea phytochemicals. Green lines represent attenuating effect, while red lines represent exacerbating effect. HPA, hypothalamic–pituitary–adrenal; CRH, corticotropin-releasing hormone; ACTH, adrenocorticotropin hormone; MCP-1, monocyte chemoattractant protein-1.

**Figure 3 nutrients-11-01361-f003:**
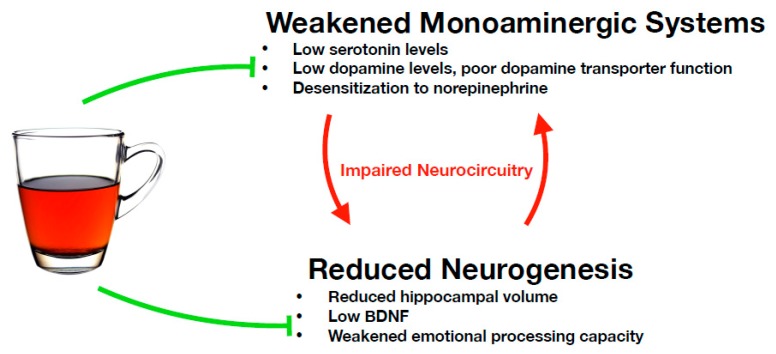
The positive feedback loop of weakened monoaminergic systems and reduced neurogenesis. Weakened monoaminergic systems and reduced neurogenesis represent two mechanistic nodes within an interrelated system of depression pathology. Bullet points identify symptoms of weakened functional status, and key targets for tea phytochemicals. Green lines represent attenuating effect, while red lines represent exacerbating effect. BDNF, brain-derived neurotrophic factor.

**Table 1 nutrients-11-01361-t001:** Tea compounds and their effects on physiological stress-response.

Tea Compound	Effect on Physiological Stress Response
L-theanine	Reduction in stress as measured by attenuated adrenal hypertrophy [31] Reduced sAA and subjective stress levels [32] Reduction in elevated plasma ACTH and CORT [32,33]
L-arginine	Reduction in stress as measured by attenuated adrenal hypertrophy [31]
Green tea polyphenol mixture	Reduced CORT and ACTH, reduced immobility in FST and TST [34] Restored HPA activity via ERK upregulation [35]
EGCG (anti-stress effects)	Restored HPA activity via ERK upregulation [35] Lowered corticosterone/CRH/ACTH [36] stress-reduction via reduction of neuron over-excitation [37,38,39] improved GABA transmission via activation of the SIRT1/PGC-1α pathway [40] Antagonized stress-reduction effects of L-theanine/L-arginine [31]
Epigallocatechin	Restored anti-stress effects of L-theanine/L-arginine [31] through competitive inhibition of EGCG [31]
Low-Caffeine Green Tea	Lowered sAA levels, improved sleep quality more effectively than standard caffeinated green tea [41] Reduction in adrenal hypertrophic stress response, more significant than standard green tea [31]
Green Tea, Shade-Grown White Tea	Lowered CgA levels following stress load task for both tea types, more effect for Shade-Grown White Tea [42]
Green Tea Aroma	Lowered CgA levels following stress load task [43]
Black Tea Aroma	Lowered CgA levels following stress load task [44]

sAA, salivary α-amylase; ACTH, adrenocorticotropin hormone; CORT, glucocorticoids; FST, forced swimming test; TST, tail suspension test; HPA, hypothalamic–pituitary–adrenal; ERK, extracellular signal-regulated kinase; CRH, corticotropin-releasing factor; GABA, gamma-aminobutyric acid; SIRT1, sirtuin 1; PGC-1α, peroxisome proliferator-activated receptor gamma (PPARγ) coactivator-1α; EGCG, epigallocatechin gallate.

**Table 2 nutrients-11-01361-t002:** Various tea compounds and their anti-inflammatory effects.

Tea Compound	Effect on Inflammatory Response
Theaflavins (TF)	Reduced LPS-induced neural inflammation, suppressed cytokine production, reduced immobility in TST. TF showed better anti-inflammatory capacity than common polyphenols, but comparable to EGCG [91].
Theaflavin-3,3′-digallate (TF3)	Inhibited LPS-induced expression TNF-α, IL-1β, and IL-6 [92,93]
Green Tea Extract	Reduced hepatic inflammation by attenuating NFκB activation via down-regulation on TNFR1 and TLR4 in HFD model [94]
Epigallocatechin gallate (EGCG)	Reduced neuroinflammation via inhibition of MAPK and NFκB pathways in HFFD model [95]. Reduced SPS-induced increase in IL-1β and TNF-α in mouse hippocampi, decreased expression level of IL-1β mRNA with RT-PCR analysis [36]. Downstream inhibitor of inflammatory signaling through occupation of TAK1 site, inhibition of p38 and NFκB [96]. Reduced CSM-induced IL-8 production via inhibition of p38, MAPK and NFκB in AC16 cardiomyocytes [97]. Inhibited phosphorylation of p65, showed in vitro NFκB regulation [98]. Inhibited activity of NO, COX-2, TNF-α, IL-6 and IL-1β in LPS-induced murine macrophage cell line [99] In cultured human epidermal keratinocytes exposed to airborne PM10, suppressed TNF-α, IL-1β, IL-6, IL-8 and MMP-1 [100]. Topical application in 8-week RCT improved acne via suppression of NFκB/AP-1 pathway [101]
EGC and EC	Effective downstream inhibitors of inflammatory signaling through occupation of TAK1 site [96]
Gallic acid	Reduced airway inflammation by decreasing IL-4, IL-5, IL-13, IL-17 in nasal lavage fluid of mice with allergic rhinitis [102]
Gallocatechin gallate	Inhibited LPS-induced expression of MCP-1 and IL-6 as effectively as EGCG in 3T3-L1 cells [98].
Oolong tea ethanol extract	Inhibited activity of NO, COX-2, TNF-α, IL-6 and IL-1β in LPS-induced murine macrophage cell line [99]
L-theanine	Topically delivered, reduced skin inflammation via inhibition of IL-1β, TNF-α and COX-2 [103]. Alleviated airway inflammation via suppression of NFκB pathway, reduced production of MCP-1, IL-4, IL-5, IL-13, TNF-α and interferon-gamma, attenuated trafficking of inflammatory cells into bronchoalveolar lavage fluid [104].
Teasaponin	Attenuated TLR-4, NF-κB, IL-1β, IL-6 and TNF-α in HFD mouse model [105].

NO, nitric oxide; COX-2, cyclooxygenase-2; TNF-α, tumor necrosis factor alpha; IL, interleukin; LPS, lipopolysaccharide; NFκB, nuclear factor kappa-light-chain-enhancer of activated B cells; RT-PCR, reverse transcription polymerase chain reaction; MCP-1, monocyte chemoattractant protein-1; HFD, high-fat diet; EGCG, epigallocatechin gallate; MAPK, mitogen-activated protein kinase; TNFR1, TNF receptor 1; TLR4, Toll-like receptor 4; HFFD, high-fat/high-fructose diet; EGC, epigallocatechin; EC, epicatechin.

**Table 3 nutrients-11-01361-t003:** Various tea compounds and their effects on monoaminergic systems.

Tea Compound	Effect on Monoaminergic Systems
L-Theanine	Increased levels of 5-HT, NE and DA in the PFC, NAC, and HIP. Increased levels of 5-HT and DA in the ST. Increased DA levels in the HIP [111]. Promoted DA transmission in HIP via DA D1/5 receptor-PKA pathway activation in AD mouse model [112]
Green tea	Administration for 5 weeks, tone of 5-HT was normalized, reduced stress response [113].
Epicatechin	Functioned as anxiolytic in OF and EPM via decreased expression of MAO-A in cortex, and increased pro-BDNF and BDNF via Akt pathway [114].
Tea Polyphenol Mixture	Reversed CUMS-induced reduction in 5-HT and NE in the HIP, PFC [115]. Partial restoration of normal DA and 5-HT following stress-induced neural injury [35]. Null effect on human GM in 12-week RCT [116]. BTP supplementation more than doubled butyric acid levels. Significant increases were observed for GTP, but to a lesser extent than BTP. Increases in SCFA upregulated AMPK activation by 70% and 289% for GTP and BTP, respectively [117].
EGCG	Partial restoration of normal DA and 5-HT following stress-induced neural injury [35]. Promoted growth of *Bifidobacterium* spp. and *Lactobacillus/Enterococcus* groups, induced higher concentrations of SCFA when incubated with human GM [118]. Enhanced adhesion of certain *Lactobacillus* strains to human epithelial intestinal lines [119].
Theaflavins	Increased DA turnover in FC, as measured by increased DOPAC and DOPAC/DA ratio [120]. Reduced oxidative stress, preserved DA levels in ST via and protection of dopaminergic neurons against degeneration by MPTP, improving motor behavior and expression of DAT and VMAT2 in ST and substantia nigra [121].
Catechin	Significantly increased *Bifidobacterium* spp. following in vitro incubation with human fecal samples [122].
EGCG3”Me	Significantly increased *Bifidobacterium* spp. and SCFA-generating GM species in vivo in mouse model [123]. Promoted growth of *Bifidobacterium* spp. and *Lactobacillus/Enterococcus* groups, induced higher concentrations of SCFA when incubated with human GM [118].
Gallocatechin gallate	Promoted growth of *Bifidobacterium* spp. and *Lactobacillus/Enterococcus* groups, induced higher concentrations of SCFA when incubated with human GM [118].
Epigallocatechin	Enhanced adhesion of certain *Lactobacillus* strains to human epithelial intestinal lines [119].

BTP, black tea polyphenol mixture; GTP, green tea polyphenol mixture; EGCG, epigallocatechin gallate; EGCG3”Me, O-Methylated EGCG; 5-HT, serotonin; NE, norepinephrine; DA, dopamine; PFC, prefrontal cortex; FC, frontal cortex; NAC, nucleus accumbens; HIP, hippocampus; PKA; AD, Alzheimer’s disease; OF, open field; EPM, elevated plus maze; MAO, monoamine oxidase; Akt, protein kinase B; CUMS, chronic unpredictable mild stress; SCFA, short-chain fatty acids; AMPK, 5′adenosylmonophosphate-activated protein kinase; GM, gut microbiota; DOPAC, 3,4-dihydroxyphenylacetic acid; DAT, dopamine transporter; VMAT2, vesicular monoamine transporter 2; MPTP, 1-methyl-4-phenyl-1,2,3,6-tetrahydropyridine.

**Table 4 nutrients-11-01361-t004:** Various tea compounds and their effects on BDNF, neurogenesis and neuroplasticity.

Tea Compound	Effect on Neurogenesis/Neuroplasticity
Teasaponin	Six-week supplementation in HFD model attenuated BDNF deficits in the HIP, prevented recognition memory impairment [105].
L-theanine	Increased exploratory activity in OFT, enhanced object recognition memory, significantly increased BDNF levels and BrdU-, Ki67, and DCX-labeled cells in the granule cell layer [188].
EGCG	DCX-positive neurons showed more elaborate dendritic trees, accompanied by significantly increased HIP neurogenesis [189]. Attenuated HFFD-induced neuronal damage, reduced cognitive disorder via upregulation of CREB/BDNF pathway [95]. Submicromolar concentrations potentiated the neuritogenic ability of BDNF in PC12 cells [190]. Attenuated corticosterone-induced cytotoxicity, up-regulated Shh pathway [191]. Induced SH-SY5Y cell growth in vitro [192]. RT-PCR showed enhanced BDNF and TrkB mRNA levels, activated Akt and CREB/BDNF pathways, inhibited sevoflurane-induced neurodegeneration, improved learning and memory retention [193]. Targeted BDNF and proBDNF signaling pathways, normalized tat-mediated increases in proapoptotic proBDNF, normalized tat-mediated decreases in mature BDNF protein in hippocampal neurons [194]. Following spinal cord injury in rats, increased expression of BDNF, GDNF, improved locomotor recovery [195].
Epigallocatechin	Enhanced the neurogenic properties of EGCG [190]. Primary EGC metabolite, EGC-M5, and several conjugated forms improved neurite length and neurite number in SH-SY5Y cells [196].
Epicatechin	Enhanced the neurogenic properties of EGCG [190]. Targeted BDNF and proBDNF signaling pathways, normalized tat-mediated increases in proapoptotic proBDNF, normalized tat-mediated decreases in mature BDNF protein in hippocampal neurons [194].
Non-EGCG GTP	Plastic changes in dendritic arborizations of dentate granule cells, improved spatial learning in Morris water maze [197].
Gallic acid	Ameliorated TMT-induced anxiety and depression, improved cell densities in the CA1, CA2, CA3 and DG hippocampal subdivisions [198].

HFD, high-fat diet; BDNF, brain-derived neurotrophic factor; HIP, hippocampus; OFT, open-field test; BrdU, 5-Bromo-2′-deoxyuridine; DCX, doublecortin; HFFD, high-fat/high-fructose; Shh, Sonic hedgehog; CREB, cyclic AMP response element binding protein; RT-PCR, reverse transcription polymerase chain reaction; TrkB, tropomyosin receptor kinase B; Akt, protein kinase B; GDNF, glial cell line-derived neurotrophic factor; EGCG, epigallocatechin gallate; GTP, green tea polyphenol mixture.

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
