# Peer review of "Mechanisms Underlying the Anti-Depressive Effects of Regular Tea Consumption"

_nutrients, 2019, doi:10.3390/nu11061361_

Reviewer 1 Report

The manuscript entitled “Mechanisms underlying the anti-depressive effect or regular tea consumption” is a comprehensive review of the different physiological procedures affecting mood and the ways that tea components can interfere. In such multidisciplinary field, I think this kind of revisions, even if long and dense, can be of help for any non-physiologist that wants to enter into this research issue.

However, because of this, it is important to be very exquisite and accurate with the description of physiological processes and systems. Sometimes I feel they are not clear, leading to some confusion. Some examples at the beginning of the manuscript are the following:

1.     In line 180, authors talk about the adrenal glands as if they were part of the kidneys. In fact, ACTH has no relationship with kidneys, interacting only with the adrenal glands.

2.     Line 182: norepinephrine, adrenalin, and cortisol are not released by ACTH. In fact, ACTH affects the adrenal cortex, as its name indicates, while these hormones are produced in the adrenal medulla, stimulated by the sympathetic system. Also, cortisol is a glucocorticoid (not known as “glucocorticoid”) and in fact, it is produced in humans, not in rodents (line319). In rodents, corticosterone is produced instead.

3.     Line 185: “the limbic system and the hippocampus, both of which serve…” The hippocampus is part of the limbic system; they are not two different things.

 These are just a few examples on how basic physiological concepts should be reviewed by a physiologist.

 Other minor points:

4.     It would be very helpful if some graphics or schemes were added to the manuscript so that concepts are clarified and easily followed.

5.     Acronyms should be defined the first time they appear in the text. Also, it is important to consider that many readers will be non-specialists so it would be very helpful to add a list of acronyms at the beginning of the manuscript.

6.     Reference style should be unified according to the rules of the journal. Some references seem to have no relationship with the concept indicated in the text. Please, use appropriate references for each concept.

7.     Where is section 4.3.1?

8.     Line 665: Please, check the font

9.     Line 694: Please, use “microbiota” instead of “microflora” even if the revised references do. It has been long ago that the term “microflora” got out-dated.

10. Maybe the authors would like to take the opportunity to discuss very recent works as Strandwitz et al, Nat Microbiol 2019, 4:396-403 on GABA-modulating bacteria.

11. Line 745: Coprococcus instead of Coproccus.

12. English grammar is very nice, however, there are several cases of typos or changes in the word order:

a.     Line 197: “viscous”, I think it is “vicious”

b.     Line 254: “This study found is that two tea types”

c.      Line 260: “antagonistic” instead of “antagonsitic”

d.     Line 369: a space is lacking

e.     Line 374: “contradictory results may be a result varying experimental setups”

f.      Line 416: Please remove the dot before brackets-

g.     Line 432: Please, remove “a”

h.     Line 451: “the role ERK signalling”

i.       Line 468: “compared to positive a control group”

j.       Line 522: “releases” should be plural, without s, I think.

k.     Line 561: “Four mg per day (of?) epicatechin in the water”

l.       Line 617: “1 are due (to?) its role in GABA neurotransmission”

m.   Line 985: “in order (to?) test the effects…”

Author Response

The manuscript entitled “Mechanisms underlying the anti-depressive effect or regular tea consumption” is a comprehensive review of the different physiological procedures affecting mood and the ways that tea components can interfere. In such multidisciplinary field, I think this kind of revisions, even if long and dense, can be of help for any non-physiologist that wants to enter into this research issue.

Thank you very much for your comments, and the time you have taken to carefully and thoroughly review this article. We believe your suggestions will help improve the content and readability of the paper, and we have done our best to edit the manuscript according to these constructive criticisms. 

However, because of this, it is important to be very exquisite and accurate with the description of physiological processes and systems. Sometimes I feel they are not clear, leading to some confusion. Some examples at the beginning of the manuscript are the following:

1.     In line 180, authors talk about the adrenal glands as if they were part of the kidneys. In fact, ACTH has no relationship with kidneys, interacting only with the adrenal glands.

Yes, that was poorly worded. This section has been thoroughly re-written to improve accuracy of physiological concepts and terminology.  

2.     Line 182: norepinephrine, adrenalin, and cortisol are not released by ACTH. In fact, ACTH affects the adrenal cortex, as its name indicates, while these hormones are produced in the adrenal medulla, stimulated by the sympathetic system. Also, cortisol is a glucocorticoid (not known as “glucocorticoid”) and in fact, it is produced in humans, not in rodents (line319). In rodents, corticosterone is produced instead.

The previous description of the physiological pathway of ACTH has been re-written and thoroughly examined for accuracy. Additionally, references to norepinephrine and adrenalin have been removed from this section, because they are not characteristic of HPA axis activity, but rather sympathetic autonomic nervous system activity, as you mentioned. In order to keep this introduction focused on HPA axis activity, we have narrowed the discussion to focus just on cortisol. Additionally, according to your suggestion, we reviewed the article and paid careful attention to assure that the term cortisol is not being used when reviewing data from rodent models. Corticosterone, rather than cortisol, as the primary glucocorticoid in rodents is clarified in the text.

3.     Line 185: “the limbic system and the hippocampus, both of which serve…” The hippocampus is part of the limbic system; they are not two different things.

Corrected. 

These are just a few examples on how basic physiological concepts should be reviewed by a physiologist.

Thank you for those points. A thorough review has been conducted of all physiological concepts and terminology introduced in this article in order to prevent the inclusion of any misleading or scientifically inaccurate information. 

Other minor points:

4.     It would be very helpful if some graphics or schemes were added to the manuscript so that concepts are clarified and easily followed.

An error was made in initially uploading our graphics. However, we have included three conceptual graphics that illustrate the overarching themes of the article. Furthermore, according to your suggestion and the suggestion of our other reviewer, we have created four tables to provide a summary of the main compounds and their observed effects within the several mechanistic ‘nodes’ we have described in the article. 

5.     Acronyms should be defined the first time they appear in the text. Also, it is important to consider that many readers will be non-specialists so it would be very helpful to add a list of acronyms at the beginning of the manuscript.

A list of acronyms has been compiled and added to the beginning of the manuscript. 

6.     Reference style should be unified according to the rules of the journal. Some references seem to have no relationship with the concept indicated in the text. Please, use appropriate references for each concept.

References have been double-checked to assure they are appropriate for each concept.

7.     Where is section 4.3.1?

Corrected

8.     Line 665: Please, check the font

Corrected

9.     Line 694: Please, use “microbiota” instead of “microflora” even if the revised references do. It has been long ago that the term “microflora” got out-dated.

Two references to “microflora” have been corrected

10. Maybe the authors would like to take the opportunity to discuss very recent works as Strandwitz et al, Nat Microbiol 2019, 4:396-403 on GABA-modulating bacteria.

In response to your suggestion, a careful analysis was done on this study and included into the section discussing gut-brain activity.  

11. Line 745: Coprococcus instead of Coproccus.

Corrected

12. English grammar is very nice, however, there are several cases of typos or changes in the word order:

a.     Line 197: “viscous”, I think it is “vicious”

Corrected

b.     Line 254: “This study found is that two tea types”

Corrected

c.      Line 260: “antagonistic” instead of “antagonsitic”

Corrected

d.     Line 369: a space is lacking

Corrected

e.     Line 374: “contradictory results may be a result varying experimental setups”

This section was removed due to your comments regarding cortisol vs corticosterone. We realized that this study was the primary piece of inconsistent data in our analysis, and it was an experiment that measured cortisol, not corticosterone, levels in rodents. For that reason it seemed appropriate to remove the study from our analysis altogether, which we believe helped the validity and consistency or our research. So, we thank you for your comment on this topic. 

f.      Line 416: Please remove the dot before brackets-

Corrected

g.     Line 432: Please, remove “a”

Corrected

h.     Line 451: “the role ERK signalling”

Corrected

i.       Line 468: “compared to positive a control group”

Corrected

j.       Line 522: “releases” should be plural, without s, I think.

Corrected

k.     Line 561: “Four mg per day (of?) epicatechin in the water”

Corrected

l.       Line 617: “1 are due (to?) its role in GABA neurotransmission”

Corrected

m.   Line 985: “in order (to?) test the effects…”

Corrected

Reviewer 2 Report

This is a very interesting review of the literature, and given the popularity of tea drinking globally, as well as the wide prevalence of depression with its associated risks, it is worthwhile to evaluate the potential of tea and tea components to combat depression. Overall the article is very well-written and the authors are very thorough in their approach. I have a few comments where improvements might be made:

1.       The authors should consider the content of the systematic review published by Grosso et al (2016) in Molecular Nutrition & Food Research on the protective effects of tea, coffee and caffeine in depression.

2.       The authors approach the mechanisms of tea in terms of 4 nodes or mechanisms. While this approach is very helpful, for clarity I suggest that the authors list out the 4 nodes in the introductory paragraph of that section (lines 159 – 173). Also, the numbering of the sections here could be improved – 4.3.2 should probably be labelled as 4.3.1, perhaps the SCFA could be incorporated into the gut-brain axis of 4.3.3. Also, clarify that the subsections within 4.3 are mechanisms by which monoaminergic systems may be modulated.

3.       Page 6: The authors describe a study in which EGCG counteracts the anti-stress effects of L-theanine, and they also describe other studies where EGCG has been shown to have effects that may be protective in depression. The authors need to make an effort to consolidate the information or to comment on what the weight of the evidence would indicate in relation to the overall effect of EGCG. Also, it may be possible to reduce some of the detail in this section to make it more concise and comprehensive.

4.       Section 4.5 (Line 863 onwards): The tone of this section seems somewhat out of place with the rest of the article. In lines 939 – 941, the authors are considering the mood-enhancing effects of tea as being separate from the effects on the HPA axis, inflammation etc. However, is it not possible, and perhaps even likely, that any mood-enhancing effects of the rituals around tea-making may be mediated by modulation of the above mechanisms via neurological interactions within the limbic system and elsewhere… a ‘Pavlov’s dog’ type of physiological phenomenon?

5.       There is a large amount of information in this manuscript. To enhance clarity and readability, it may be worthwhile using some tables to summarise information eg. Using a table for each ‘node’ to summarise effects of tea components on that mechanism.

6.       Line 917 – 919: reference needed

7.       There are a number of minor typos and errors throughout the manuscript. Here are some of them:

a.       Line 198: viscious

b.      Line 254: found that

c.       Line 260: antagonistic

d.      Line 374: result of

e.      Line 432: ‘a’ Lee et al

f.        Line 468: compared to a positive control group

g.       Line 589: striatum

h.      Line 767: striatum

Author Response

This is a very interesting review of the literature, and given the popularity of tea drinking globally, as well as the wide prevalence of depression with its associated risks, it is worthwhile to evaluate the potential of tea and tea components to combat depression. Overall the article is very well-written and the authors are very thorough in their approach. I have a few comments where improvements might be made:

Thank you very much for your comments, and the time you have taken to carefully and thoroughly review this article. We believe your suggestions will help improve the content and readability of the article, and we have done our best to edit the manuscript according to these constructive criticisms.

1.       The authors should consider the content of the systematic review published by Grosso et al (2016) in Molecular Nutrition & Food Research on the protective effects of tea, coffee and caffeine in depression.

In response to your comment, we thoroughly investigated the systematic review published by Grosso et al (2016). We found that there were issues with their research methodology that concerned us. Firstly, they only used 5 studies in their meta-analysis of tea and depression, 3 of which are already included in our current review within a different meta-analyses. Importantly, the other two studies, which were male and female constituencies from the same study by Guo et al. (2015), showed evidence of publication bias according to Grosso et al. This study was a clear outlier compared to the other studies analyzed by Grosso et al. and the other meta-analyses on the association of tea consumption and depression, such as that by Dong et al. (2015) that we have included in our current article currently. I have included here a quote from Grosso et al. (2015) about their meta-analysis on tea and depression; 

“Pooled analysis of both cross-sectional and prospective studies showed a borderline nonsignificant association between tea consumption and risk of depression (RR: 0.70, 95% CI: 0.48, 1.01; Fig. 3), with significant heterogeneity and suspect of publication bias at funnel plot (Supporting Information Fig. 2). In fact, both forest and funnel plot demonstrated heterogeneity due to the results of the study of Guo et al. [24], which reported nonsignificant increased risk of depression. The study had the largest sample size accounting for the highest weight in our models. Sensitivity and subgroup analyses confirmed that exclusion of this study resulted in a significant reduced risk of depression by tea consumption (RR 0.50, 95% CI: 0.38, 0.66)”

We examined the funnel plot in Supporting Information Fig. 2 and it is apparent that this study, which represents a majority of the data analyzed by Grosso et al., does fit the criteria for publication bias, and for that reason we find this meta-analysis unfit for inclusion in our review. 

If it is insisted by the reviewer that it must be included, we will include it, and we will provide a description to our readers to interpret the data with caution for the reasons we have explained here. We hope that this has provided a satisfactory response to your comment, and we welcome any additional comments or opinions on this point. 

 2.       The authors approach the mechanisms of tea in terms of 4 nodes or mechanisms. While this approach is very helpful, for clarity I suggest that the authors list out the 4 nodes in the introductory paragraph of that section (lines 159 – 173). Also, the numbering of the sections here could be improved – 4.3.2 should probably be labelled as 4.3.1, perhaps the SCFA could be incorporated into the gut-brain axis of 4.3.3. Also, clarify that the subsections within 4.3 are mechanisms by which monoaminergic systems may be modulated.

According to your suggestion, we have listed out the 4 nodes in the second sentence of the introductory paragraph of that section. We have re-ordered the numbering of the sections to make 4.3.2 labeled as 4.3.1. Also, into the introduction section of monoaminergic health restoration we included an explanation of our inclusion of gut microbiota and SCFA into this section, so that the reader will understand that these parts are subsections of monoaminergic health.  

3.       Page 6: The authors describe a study in which EGCG counteracts the anti-stress effects of L-theanine, and they also describe other studies where EGCG has been shown to have effects that may be protective in depression. The authors need to make an effort to consolidate the information or to comment on what the weight of the evidence would indicate in relation to the overall effect of EGCG. Also, it may be possible to reduce some of the detail in this section to make it more concise and comprehensive.

Agreed. Following your suggestion, we consolidated a good deal of information and addressed the issue of EGCG and L-theanine. There are currently no mechanistic explanations in the literature of how EGCG could have counteracted the anti-stress effects of L-theanine. So, we decided to conduct a brief but thorough investigation of the literature with the limited time we had, and we were ultimately able to construct what we believe is the first tenable hypothesis of how EGCG may be functioning to antagonize the anti-stress effects of L-theanine. We believe that the inclusion of this novel analysis into the manuscript has improved it’s value to the field. We encourage you to provide your thoughts on our newly proposed explanation, and we thank you for encouraging us to make an effort to think more critically about this topic.  

4.       Section 4.5 (Line 863 onwards): The tone of this section seems somewhat out of place with the rest of the article. In lines 939 – 941, the authors are considering the mood-enhancing effects of tea as being separate from the effects on the HPA axis, inflammation etc. However, is it not possible, and perhaps even likely, that any mood-enhancing effects of the rituals around tea-making may be mediated by modulation of the above mechanisms via neurological interactions within the limbic system and elsewhere… a ‘Pavlov’s dog’ type of physiological phenomenon?

According to your comments, we have re-written most of this section in order to convey our thoughts more clearly. We agree that the previous tone of this section was somewhat mismatched with previous sections. What we believe is now expressed more clearly is that for some people, there may be a difference between preparing/drinking a cup of tea and simply taking a tea extract pill, even if both scenarios provide the subject with the same quantities of bioactive compounds. The ‘Pavlov’s dog’ sort of physiological phenomenon is a very interesting concept. As you say, it seems likely that the tea drinker may be gaining improvements to mood because of physiological conditioning of improvements to the HPA axis, inflammation, etc. However, to say that improvements to mood are only a conditioned response to physiological benefits might fall short to consider that many drinkers find pleasure in the taste of tea, the aroma, the appreciation of the leaves, the time one takes out of their busy day to sit down and rest and drink the cup of tea, the activity in it of itself. Our purpose in writing this section was to explain that these sort of factors alone might be able to enhance mood, even without the observed physiological effects resulting from phytochemicals, i.e. HPA axis, inflammation, etc. While we agree that a pavlov’s dog-like conditioning to physiological phenomena is indeed likely a relevant factor, we also believe that the experience itself, can play an important role for some people. We are sure to emphasize the variability and subjectivity of this mechanistic pathway. While it is inconsistent, we believe this has shown to be a real effect under certain circumstances.

5.       There is a large amount of information in this manuscript. To enhance clarity and readability, it may be worthwhile using some tables to summarise information eg. Using a table for each ‘node’ to summarise effects of tea components on that mechanism.

We fully agree with this point. We have made figures and charts to enhance clarity and readability. 

6.       Line 917 – 919: reference needed

That sentence was removed, and references were added to the remaining sentences in that paragraph. 

7.       There are a number of minor typos and errors throughout the manuscript. Here are some of them:

a.       Line 198: viscious

Corrected

b.      Line 254: found that

Corrected

c.       Line 260: antagonistic

Corrected

d.      Line 374: result of

Corrected

e.      Line 432: ‘a’ Lee et al

Corrected

f.        Line 468: compared to a positive control group

Corrected

g.       Line 589: striatum

Corrected

h.      Line 767: striatum

Corrected

Round  2

Reviewer 1 Report

Dear authors,

 Thank you for carefully addressing all my comments. It is a nice revision. Good luck.